# Nitrogen coordinated import and export of arginine across the yeast vacuolar membrane

**Melody Cools** [1,2], **Simon Lissoir** [1], **Elisabeth Bodo** [3], **Judith Ulloa-Calzonzin** [4], **Alexander DeLuna** [4], **Isabelle Georis** [2], **Bruno André** [1]*

**1** Molecular Physiology of the Cell, Université Libre de Bruxelles (ULB), Biopark, Gosselies, Belgium, **2** Métabolisme des micro-organismes modèles, LABIRIS, Brussels, Belgium, **3** Développement des bioprocédés et microbiologie appliquée, LABIRIS, Brussels, Belgium, **4** Unidad de Genómica Avanzada (Langebio), Centro de Investigación y de Estudios Avanzados del IPN, Irapuato, Guanajuato, Mexico

* Bruno.Andre@ulb.ac.be

**Data Availability Statement:** All relevant data are within the manuscript and its Supporting Information files.

## Abstract

The vacuole of the yeast *Saccharomyces cerevisiae* plays an important role in nutrient storage. Arginine, in particular, accumulates in the vacuole of nitrogen-replete cells and is mobilized to the cytosol under nitrogen starvation. The arginine import and export systems involved remain poorly characterized, however. Furthermore, how their activity is coordinated by nitrogen remains unknown. Here we characterize Vsb1 as a novel vacuolar membrane protein of the APC (amino acid-polyamine-organocation) transporter superfamily which, in nitrogen-replete cells, is essential to active uptake and storage of arginine into the vacuole. A shift to nitrogen starvation causes apparent inhibition of Vsb1-dependent activity and mobilization of stored vacuolar arginine to the cytosol. We further show that this arginine export involves Ypq2, a vacuolar protein homologous to the human lysosomal cationic amino acid exporter PQLC2 and whose activity is detected only in nitrogen-starved cells. Our study unravels the main arginine import and export systems of the yeast vacuole and suggests that they are inversely regulated by nitrogen.

## Author summary

The lysosome-like vacuole of the yeast *Saccharomyces cerevisiae* is an important storage compartment for diverse nutrients, including the cationic amino acid arginine, which accumulates at high concentrations in this organelle in nitrogen-replete cells. When these cells are transferred to a nitrogen-free medium, vacuolar arginine is mobilized to the cytosol, where it is used as an alternative nitrogen source to sustain growth. Although this phenomenon has been observed since the 1980s, the identity of the vacuolar transporters involved in the accumulation and the mobilization of arginine is not well established, and whether these processes are regulated according to nutritional cues remains unknown. In this study, we exploited *in vitro* and *in vivo* uptake assays in vacuoles to identify and characterize Vsb1 and Ypq2 as vacuolar membrane proteins mediating import and export of arginine, respectively. We further provide evidence that Vsb1 and Ypq2 are inversely regulated according to the nitrogen status of the cell. Our study sheds new light on the poorly

**Funding:** M.C. was the recipient of a PhD fellowship from the Fonds pour la Formation à la Recherche dans l'Industrie et dans l'Agriculture (FRIA). This work was supported by a PDR grant (nr. 746 23655065) by the Fonds National de la Recherche Scientifique (FNRS) (Fédération Wallonie-Bruxelles, Belgium), the International Brachet Foundation, and by a grant (nr. CRFF-2015-01) from the Cystinosis Research Foundation. The funders had no role in study design, data collection and analysis, decision to publish, or preparation of the manuscript.

**Competing interests:** The authors have declared that no competing interests exist.

studied topic of the diversity and metabolic control of vacuolar transporters. It also raises novel questions about the molecular mechanisms underlying their coordinated regulation and, by extension, the regulation of lysosomal transporters in human cells.

## Introduction

The vacuole of the yeast *Saccharomyces cerevisiae* is the counterpart of the lysosome and has proved to be a valuable model for studying this organelle [1]. The main role of the yeast vacuole, like that of lysosomes, is to carry out the degradation of proteins and other macromolecules delivered to it via the endocytic or the autophagic pathway. The released metabolites are then exported to the cytosol via diverse transporters [2]. In humans, dysfunction of a single lysosomal hydrolase or transporter can cause detrimental accumulation of non-recycled metabolites, the typical feature of lysosomal storage diseases (LSDs) [3]. One such disease, cystinosis, is caused by mutations in the CTNS gene encoding cystinosin, a lysosomal cystine exporter [4]. Patients suffering from cystinosis are treated with the aminothiol cysteamine. This molecule enters the lysosomes and reacts there with accumulated cystine, converting it to cysteine and a cysteine-cysteamine mixed disulfide. The latter compound, structurally similar to lysine, is then efficiently exported from the lysosomes via the PQLC2 cationic amino-acid exporter [5].

In yeast, the closest homologs of PQLC2 are the proteins Ypq1, 2 and 3. They localize to the vacuolar membrane via the ALP (alkaline phosphatase) traffic pathway [6] and are involved in homeostasis of cationic amino acids. Specifically, *ypq1Δ* and *ypq2Δ* deletion mutants display resistance to canavanine, a toxic analog of arginine (Arg), and the *YPQ3* gene is transcriptionally repressed by excess lysine (Lys). The canavanine resistance phenotype of the *ypq2Δ* mutant can be suppressed by expression of PQLC2, shown to also recognize the Arg analog, suggesting that Ypq2 likewise functions as a vacuolar exporter of cationic amino acids [5]. Since this discovery, however, investigators have reported experiments on reconstituted vacuolar vesicles showing that Ypq1 and Ypq3 respectively catalyze Lys and histidine (His) uptake into the vacuole and that their activity depends on the $H^+$ gradient established by the V-ATPase [7,8], while Ypq2 catalyzes Arg/His exchange [9]. Another study reported that Ypq1, under Lys starvation conditions, is targeted to the vacuolar lumen and degraded, a result supporting a role of Ypq1 in Lys uptake into the vacuole [10,11].

Transport of cationic amino acids, and particularly Arg, into the yeast vacuole has been investigated in other previous studies. For instance, the pioneering works of A. Wiemken and collaborators, based on assays of transport into intact vacuoles, revealed the existence of a high-affinity transporter ($K_m$ ~30 μM) catalyzing Arg/Arg exchange with 1:1 stoichiometry [12]. The corresponding activity was measurable even in the absence of ATP (and hence of the $H^+$ gradient across the membrane). The molecular identity of this transport protein, however, has remained unknown. Later, the group of Y. Anraku reported that the proteins Vba1, 2 and 3 function as secondary active importers of basic amino acids into reconstituted yeast vacuolar vesicles. The Arg transport activity measured under these conditions is characterized by a lower affinity ($K_m$ ~0.65 mM) [13]. More recently, Avt4 has been reported to function as a secondary active exporter of basic amino acids [14] and, depending on the experimental conditions, the Avt1 neutral amino acid transporter also seems able to import His [15,16]. During finalization of this manuscript, Ypq2 was reported to catalyze Arg uptake coupled to His efflux [9]. Overall, the respective roles and relative importance of transporters mediating influx and efflux of basic amino acids across the yeast vacuolar membrane remain poorly understood.

That the yeast vacuole is the major storage compartment for Arg and other cationic amino acids is well established [17,18]. High intravacuolar Arg concentrations are achieved through interactions between positive charges on the cationic amino acid and negatively charged poly-phosphate chains [19]. The latter are synthesized by the vacuolar transporter chaperone (VTC) complex [20] and accumulate in the vacuole at high concentration (~ 100 mM phosphate groups) [19,21,22]. This interaction between Arg and polyphosphates presumably results in efficient and energy-cost-effective sequestration of both compounds in vacuoles, without preventing their independent mobilization according to varying growth conditions [19].

This study aims to further characterize the molecular determinants of Arg transport (export and import) across the yeast vacuolar membrane and to assess whether they are under nitrogen control. We report that Ypq2, the yeast homolog of human PQLC2, plays a particularly important role in recycling vacuolar Arg stores to the cytosol under nitrogen starvation. We further characterize Vsb1/Ygr125w as a novel vacuolar membrane protein playing an essential role in accumulation of Arg and other cationic amino acids within the vacuole. Finally, we provide evidence that Ypq2 and Vsb1 are inversely regulated according to the nitrogen supply conditions.

## Results

### Conditions for assaying amino acid transport into intact vacuoles

To study the transport of amino acids into and out of the yeast vacuole, we considered isolated intact vacuoles to be the most appropriate material, as they retain their sap and metabolite content, including amino acids, and should be more similar to the vacuoles naturally present in cells than the reconstituted vacuolar vesicles used in many previous works. Our first step was thus to implement and adapt methods for isolating intact vacuoles and measuring the uptake of radiolabeled amino acids into them. To assess the purity and integrity of our isolated vacuoles and to readily measure their internal pH, we isolated cells stably expressing the vacuolar lumen protein Sna3 fused to pHluorin, the vacuolar membrane protein Sna4 fused to DsRed, or both (Fig 1A). Microscopy analysis of cells expressing both constructs showed, as expected, localization of Sna3-pHluorin to the lumen and of Sna4-DsRed to the peripheral membrane of the vacuole (Fig 1B) [23,24]. The method of Cabrera and Ungermann for isolating intact vacuoles was then applied to these cells [25]. Isolated DsRed-labeled vacuoles appeared intact, as all of them also displayed luminal pHluorin labeling, i.e. they obviously had not ruptured and should have conserved their sap (Fig 1B). Vacuoles labeled only with Sna3-pHluorin were then isolated and incubated with the lipophilic red fluorescent dye FM4-64, which should stain all isolated membranes. All the isolated compartments (stained in red) were found to display luminal pHluorin green fluorescence, confirming that the isolated compartments visible under the microscope were vacuoles only (Fig 1B).

Taking advantage of the fact that pHluorin enables ratiometric measurement of the pH, we investigated whether the V-ATPase of the isolated vacuoles was active. Several observations suggested that this was the case: the pH measured in isolated vacuoles became more acidic in the presence of ATP (Fig 1C), this acidification was partially lost upon addition of the $K^+/H^+$ ionophore nigericin, and it was not observed if the vacuoles were pre-incubated with the V-ATPase inhibitor bafilomycin A (Fig 1C). The final pH of the isolated vacuoles after ATP addition (~6.2) was in the upper part of the *in vivo* pH range of the vacuole [26].

The vacuolar Avt1 transporter is reported to catalyze tyrosine uptake. This transport activity, measured on vacuolar vesicles, depends on the $H^+$ gradient established by the V-ATPase [15]. Using intact vacuoles instead of vacuolar vesicles, we likewise measured ATP-dependent tyrosine uptake (Fig 1D). This uptake activity was lost if nigericin or a V-ATPase inhibitor was

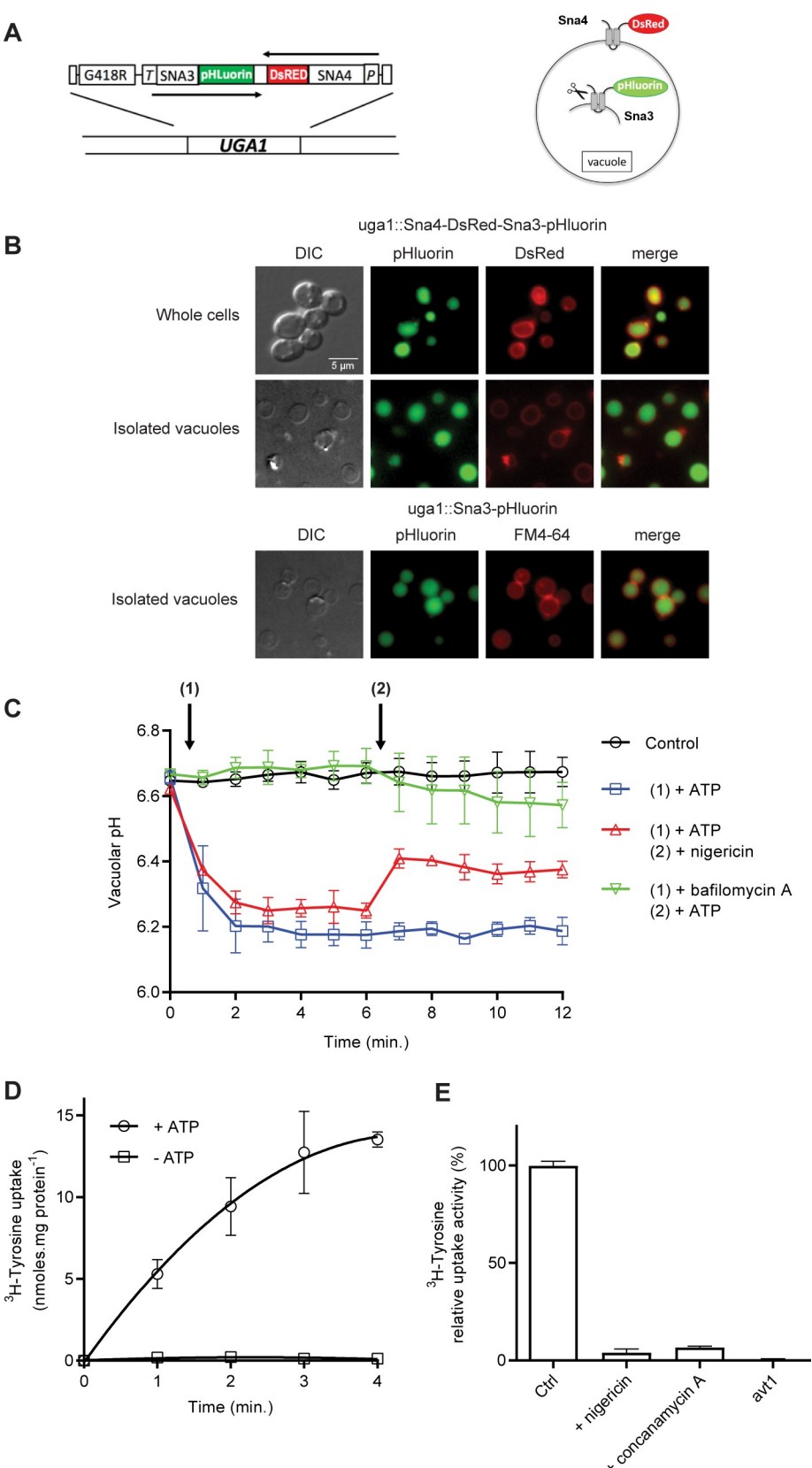

**Fig 1. Isolation of intact vacuoles suitable for amino acid transport assays.** *(A)* Left. Sketch of the DNA cassette integrated into the *UGA1* locus and expressing the *SNA3-PHLUORIN* and *SNA4-DsRED* hybrid genes under the control of the *TPI1* (T) or *PGK1* (P) gene promoter, respectively. Strains having integrated this cassette were selected for resistance to geneticin (G418R) and loss of the ability to use GABA as a nitrogen source (*uga1Δ* phenotype). Right. Schematic representation of Sna3-pHluorin and Sna4-DsRed located, respectively, at the peripheral membrane and in the lumen of the vacuole. Sna3 is a transmembrane protein targeted to the vacuole via the multivesicular-body (MVB) pathway. Scissors represent vacuolar proteases. *(B)* Cells expressing Sna3-pHluorin and/or Sna4-DsRed and samples of vacuoles isolated from them were examined by epifluorescence microscopy. In whole cells, Sna3-pHluorin and Sna4-DsRed correctly label the vacuolar lumen and limiting membrane, respectively. Isolated vacuoles exhibit the same labeling. Vacuoles isolated from cells only expressing Sna3-pHluorin were labeled with FM4-64 (10 μM). *(C)* The V-ATPase is active in isolated vacuoles. The pH of vacuoles isolated from cells expressing Sna3-pHluorin was measured after addition of ATP (4 mM), nigericin (6.5 μM), or bafilomycin A (6.5 μM) (n = 2). *(D)* Time course of [$^3$H]-L-tyrosine (50 μM) uptake into vacuoles isolated from the *w-t* strain. Vacuoles were pre-incubated for 8 minutes with or without ATP (4 mM) before adding the radiolabeled amino acid (n = 2). *(E)* Accumulation of $^3$H-tyrosine (50 μM) was measured in vacuoles from the *w-t* or the *avt1Δ* mutant after a 4-minute incubation. The vacuoles were pre-incubated with ATP (4 mM), and with or without nigericin (6.5 μM) or concanamycin A (6.5 μM) (n = 4). For all experiments, error bars represent the SD.

also added to the transport reaction (Fig 1E). Nor was it detected in vacuoles isolated from the *avt1Δ* mutant (Fig 1E).

In conclusion, the intact yeast vacuoles isolated via our method have a functional V-ATPase and appear well suited for measuring the activity of amino acid transporters.

## Arginine uptake into isolated vacuoles depends mainly on Ypq2

We next used the above-described transport assay to measure the uptake of Arg (50 μM) into vacuoles. In the presence of ATP, the measured arginine transport activity was of the same order of magnitude as tyrosine transport activity (Fig 2A). In the absence of ATP, it was reduced much less markedly than tyrosine uptake, to about half the level observed in the presence of ATP. A similar reduction in Arg transport activity was observed in the presence of ATP, upon concomitant addition of a protonophore such as FCCP or nigericin and when the vacuoles were incubated with the V-ATPase inhibitor concanamycin A (Fig 2B). The K$^+$ ionophore valinomycin did not significantly affect Arg uptake. The H$^+$ gradient established by the V-ATPase thus appears to stimulate Arg uptake.

Previous experiments on reconstituted vacuolar vesicles have shown several transporters to contribute to Arg transport across the vacuolar membrane (see Introduction). We thus tested their contribution to Arg uptake in intact vacuoles and found that practically no uptake at 50 μM is detectable in vacuoles isolated from *ypq2Δ* mutant cells, both in the presence or absence of ATP (Fig 2C). Ypq2 is thus the main Arg transport system detectable in isolated vacuoles, and its activity is somehow stimulated when an H$^+$ gradient is established by the V-ATPase.

We next measured the uptake of Arg added at higher concentrations (Fig 2D). In vacuoles of the *w-t* strain, uptake increased, as expected, when the Arg concentration rose, and the stimulating effect of ATP was observed at all tested concentrations. Arg uptake was greatly reduced in vacuoles isolated from the *ypq2Δ* mutant strain, but residual uptake, not stimulated by ATP, remained detectable when Arg was added at high concentrations. We tested whether this residual Arg uptake might involve the vacuolar protein Ypq1 or Ypq3, both of which are similar in sequence to Ypq2 [5]. Vacuoles from the *ypq1-2-3Δ* triple mutant showed lower uptake than vacuoles from the *ypq2Δ* single mutant, but a residual uptake not stimulated by ATP was still detected.

In conclusion, in our *in vitro* assay of Arg transport into intact vacuoles, Ypq2 behaves as the highest-affinity uptake system, and its activity is stimulated by ATP. It seems that Ypq1, Ypq3, or both also contribute to Arg uptake, though apparently with lower affinity.

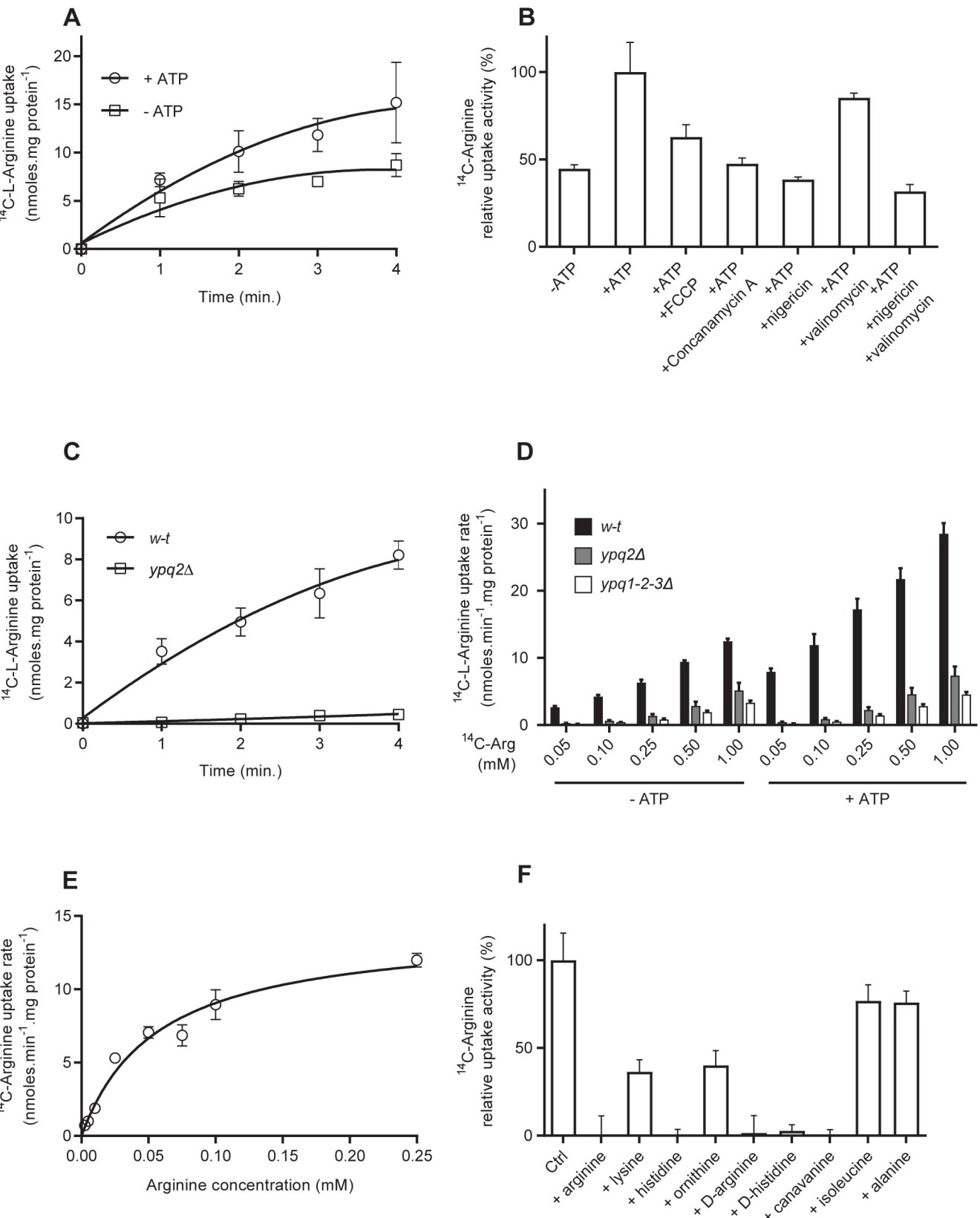

**Fig 2. Arg uptake into isolated vacuoles depends mainly on Ypq2.** *(A)* Time course of $^{14}$C-Arg (50 μM) uptake into intact vacuoles from the *w-t*. The vacuoles were incubated for 8 minutes in the absence or presence of ATP (4 mM) before addition of $^{14}$C-Arg (n = 3). *(B)* Sensitivity to inhibitors and ionophores. Accumulation of $^{14}$C-Arg (50 μM) was measured in vacuoles after a 4-minute incubation. V-ATPase inhibitors (6.5 μM) and ionophores (6.5 μM) were added simultaneously with $^{14}$C-Arg (n = 3). *(C)* Time course of $^{14}$C-Arg uptake, in the absence of ATP, into intact vacuoles from the *w-t* and *ypq2Δ* strains (n = 2–3). *(D)* Accumulation of $^{14}$C-Arg added at different concentrations (mM) was measured after a 4-minute incubation in vacuoles isolated from the *w-t*, *ypq2Δ*, and *ypq1-2-3Δ* strains. The vacuoles were incubated for 8 minutes in the absence and presence of ATP (4 mM) before addition of $^{14}$C-Arg (n = 3). *(E)* Saturation kinetics of Arg uptake through Ypq2 into isolated vacuoles. Uptake into vacuoles of $^{14}$C-Arg added at different concentrations was measured after a 30-second incubation in the absence of ATP. Values for the *ypq1-2-3Δ* strain were substracted from those for the *ypq1-3Δ* strain. $K_m$ = 56 ± 10 μM, $V_{max}$ = 14.18 ± 0.06 nmoles.mg prot$^{-1}$.min$^{-1}$ (n = 2). *(F)* Substrate selectivity of Ypq2. Accumulation of $^{14}$C-Arg (50 μM) in vacuoles was measured, in the absence of ATP, after a 4-minute incubation (see text for the strains used). Inhibitors (5 mM) were added simultaneously with $^{14}$C-Arg (n = 2). For all experiments, error bars represent the SD.

Furthermore, detection of residual uptake into vacuoles from the *ypq1-2-3Δ* triple mutant suggests that yet another Arg transport system, displaying even lower affinity, is present at the vacuolar membrane.

As Ypq2 is the highest-affinity Arg transporter detected in isolated vacuoles, we measured its apparent $K_m$. To specifically measure the Ypq2-dependent activity, we assayed the uptake of Arg added at different concentrations in the *ypq1-3Δ* and *ypq1-2-3Δ* mutant strains, and subtracted the values obtained for the triple-mutant vacuoles from those obtained for the double-mutant vacuoles. The $K_m$ determined was 56 ± 10 μM and the $V_{max}$ was 14.18 ± 0.06 nmoles.mg prot$^{-1}$.min$^{-1}$ (Fig 2E).

The same strategy was applied to study Ypq2 selectivity (Fig 2F). Different amino acids and analogs were tested for their ability to inhibit Ypq2-dependent Arg uptake. As expected, non-radiolabeled Arg inhibited the uptake of $^{14}$C-Arg, and two tested neutral amino acids (isoleucine and alanine) did not significantly inhibit this uptake. Canavanine, the D-Arg and the L- and D-isomers of His completely inhibited the uptake, whereas Lys or ornithine addition led to ~50% inhibition of uptake. We thus tested whether Ypq2 contributes also to His and Lys uptake. His uptake into vacuoles from *the ypq2Δ* mutant was found to be strongly reduced, but Lys uptake was not (even when Lys was present at 1 mM) (S1 Fig).

In conclusion, the main high-affinity Arg uptake activity detected in isolated vacuoles is due to Ypq2. This transport protein is more active when the H$^+$ gradient is established by the V-ATPase. It is also capable of catalyzing His but not Lys uptake.

## Ypq2-dependent uptake of arginine into isolated vacuoles is primarily an exchange reaction

A. Wiemken and collaborators [12,27] previously detected in intact vacuoles an uptake of radiolabeled Arg displaying features very similar to those mentioned above for Ypq2: it was not ATP dependent but was stimulated about 2-fold by ATP; it showed relatively high affinity (a calculated $K_m$ of about 30 μM) close to our estimate; it was inhibited by D-Arg, His, and canavanine. These facts suggest that Ypq2 is the transport system responsible for the Arg uptake activity described by Wiemken's group. Importantly, this group also reported that the system catalyzes stoichiometric Arg exchange (ratio 1:1). To further investigate this matter, we examined the influence of the luminal Arg concentration on Ypq2-dependent uptake of $^{14}$C-Arg. For this we isolated intact vacuoles and caused their osmotic rupture so as to generate small vacuolar vesicles [28]. The quality of these vesicles was first tested by measuring tyrosine uptake. As described above for intact vacuoles, the vesicles displayed Tyr uptake strongly stimulated by ATP, and when they were obtained from *avt1Δ* mutant cells, this uptake was markedly reduced (Fig 3A). We next isolated vesicles from the *w-t* and *ypq2Δ* mutant strains. Vesicle formation was carried out in the absence or presence of a high Arg concentration, in the same range as that measured in the vacuole [17,29]. These reconstituted vesicles, filled or

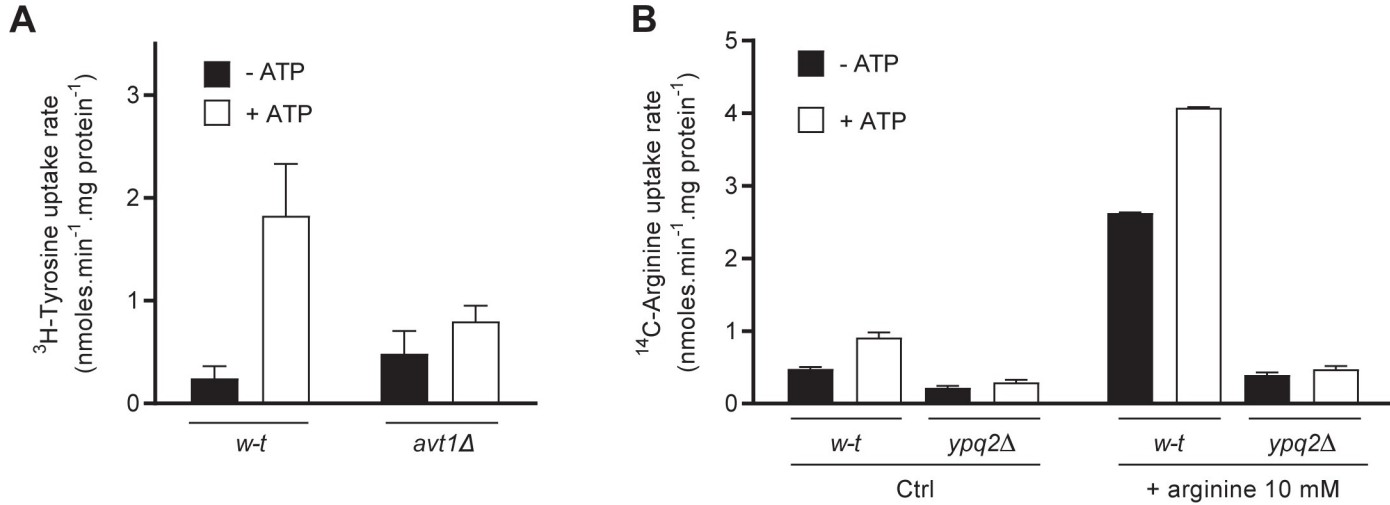

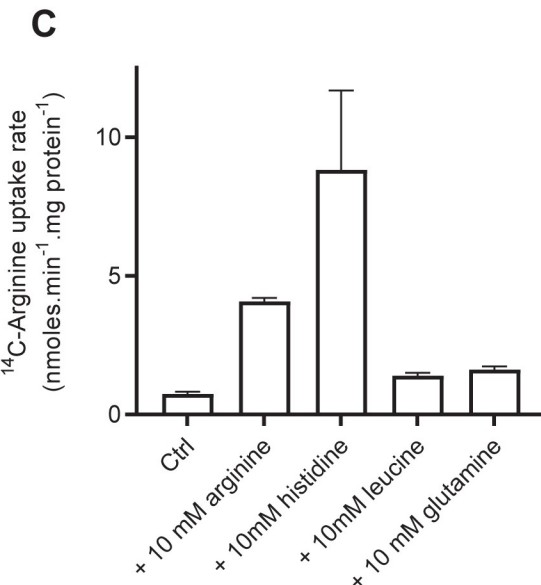

**Fig 3. Ypq2 catalyzes an Arg/Arg exchange reaction.** *(A)* Accumulation of $^3$H-tyrosine (50 μM) after a 4-minute incubation was measured in reconstituted vesicles prepared from vacuoles of the *w-t* and *avt1Δ* mutant strains. Vesicles were incubated for 8 minutes in the absence and presence of ATP (4 mM) before addition of $^3$H-tyrosine (n = 2). *(B)* The uptake of $^{14}$C-Arg added at 50 μM was measured, after a 4-minute incubation, in reconstituted vesicles prepared from vacuoles of the *w-t* and *ypq2Δ* mutant strains. Vesicles generated in a buffer without Arg (Ctrl) or containing 10 mM non-radiolabeled Arg were incubated for 8 minutes in the absence or presence of ATP (4 mM) before addition of $^{14}$C-Arg (n = 2). *(C)* The uptake of $^{14}$C-Arg added at 50 μM was measured, after a 4-minute incubation, in reconstituted vesicles prepared from vacuoles of the *w-t* strain. The vesicles were filled with a buffer without any amino acid (Ctrl) or with 10 mM non-radiolabeled Arg, His, leucine, or glutamine. The vesicles were then incubated for 8 minutes in the presence of ATP (4 mM) before addition of $^{14}$C-Arg (n = 2). For all experiments, error bars represent the SD.

not with Arg, were washed by successive centrifugations and used to measure the uptake of $^{14}$C-Arg in the absence and presence of ATP (Fig 3B). In vesicles generated without added Arg, Arg uptake was low compared to tyrosine uptake, whether ATP was present or not, although ATP stimulated the uptake. Remarkably, Arg-filled vesicles showed much higher Arg

uptake, and this uptake depended on Ypq2 and was stimulated by ATP (Fig 3B). Other vesicle-entrapped amino acids such as Leu or Gln did not support Arg uptake (Fig 3C). In contrast, loading the vesicles with His was found to drive high Arg uptake (Fig 3C). This is in agreement with a report published during finalization of this paper, showing that Ypq2-catalyzed Arg uptake into vacuole-derived vesicles is stimulated by luminal His [9].

These results indicate that Ypq2-dependent uptake of Arg into intact vacuoles is mainly an exchange reaction, and that the luminal amino acid driving the Arg transport reaction can be Arg or His.

## Ypq2 is involved in exporting intravacuolar arginine under nitrogen starvation

Unlike Avt1, Ypq2 is active in vacuoles in the absence of ATP (Fig 2A and 2B), i.e., in the absence of an $H^+$ gradient established by the V-ATPase. Appreciable Ypq2 activity does require, however, the presence of at least one of its substrates inside the vacuole, since little $^{14}C$-Arg was taken up by the vacuolar vesicles devoid of Arg or His (Fig 3C). As it seems doubtful that such an exchange reaction would be biologically relevant, we envisaged the possibility that Ypq2 might catalyze net efflux of intravacuolar Arg or His *in vivo* when the cytosolic concentrations of these amino acids are low. This could occur in nitrogen-starved cells, where cytosolic amino acids become scarce. The exchange activity measured *in vitro* might thus correspond to the situation occurring when the cytosolic concentrations of Ypq2 substrates return to levels close to the measured $K_m$ of the transporter. This model, suggesting a role for Ypq2 mainly in exporting intravacuolar Arg and His, is supported by the canavanine resistance phenotype displayed by *ypq2Δ* mutant cells, a probable consequence of sequestration of the toxic compound in the vacuole [5]. In addition, the higher Ypq2 activity measured in vacuoles acidified by the V-ATPase supports the view that, rather than functioning as a proton antiporter, the Ypq2 protein evolved to function optimally at a pH close to that of the vacuolar lumen.

In line with this model, a previous study has shown that vacuolar Arg stocks are mobilized when cells are shifted to a medium devoid of nitrogen (N) [29]. To assess the possibility that Ypq2 might play a role in Arg mobilization from the vacuole during N starvation, we measured by HPLC the intracellular Arg content of *w-t* and *ypq2Δ* mutant cells, before and 6 hours after a shift to a medium without N. We found the Arg content of *w-t* cells, known to correspond largely to their vacuolar stocks [29], to decrease upon N starvation, as expected if the stored vacuolar Arg was exported from the vacuole and used by the cells (Fig 4A). This Arg decrease was not observed in the *car1* mutant lacking arginase, the first Arg-catabolic enzyme. This indicates that the major part of the mobilized vacuolar Arg is used as an N source rather than being incorporated into newly synthesized proteins. In *car1* mutant cells shifted to N starvation conditions, the measured Arg level even rose above the initial concentration. This likely reflects mobilization of additional Arg molecules through autophagy. Interestingly, the *ypq2Δ* strain displayed after 6 h of starvation a much higher residual intracellular Arg level than the *w-t* strain, and this difference was not detected in *car1* mutant cells (Fig 4A). This result shows that Ypq2 contributes importantly to the consumption of intracellular Arg by N-starved cells, most likely by catalyzing net export of intravacuolar Arg. The residual Arg consumption observed in the *ypq2Δ* mutant could correspond to the Arg initially present in the cytosol and other organelles and/or to a contribution of additional vacuolar Arg exporters.

When catalyzing intravacuolar Arg export in N-starved cells, Ypq2 could in principle function as an exchanger, exporting Arg in exchange for another compound present in the cytosol. This cytosolic compound might in principle be His, another substrate of Ypq2, but this seems unlikely as more than 90% of the cellular His pool is stored in the vacuole of N replete cells

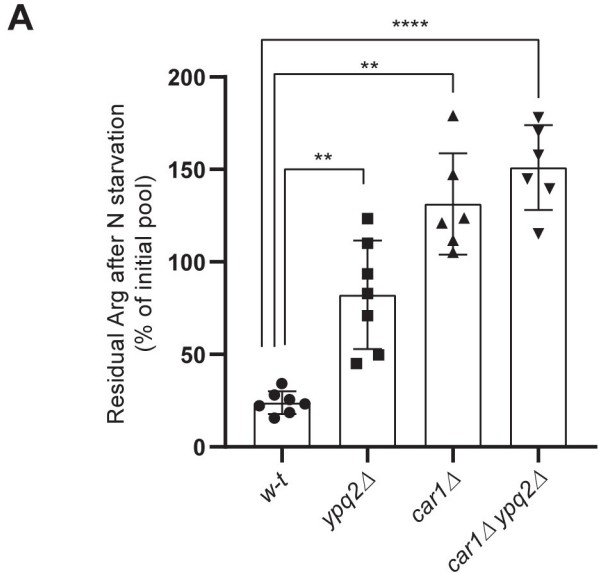

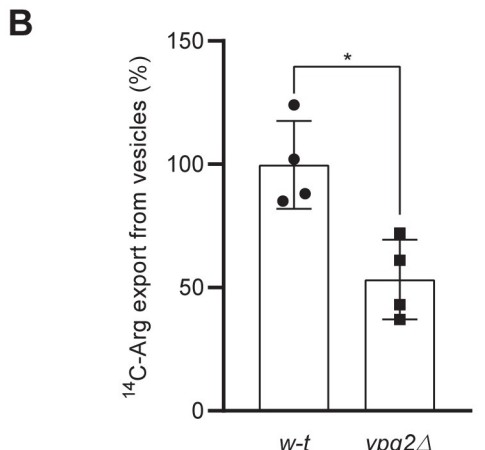

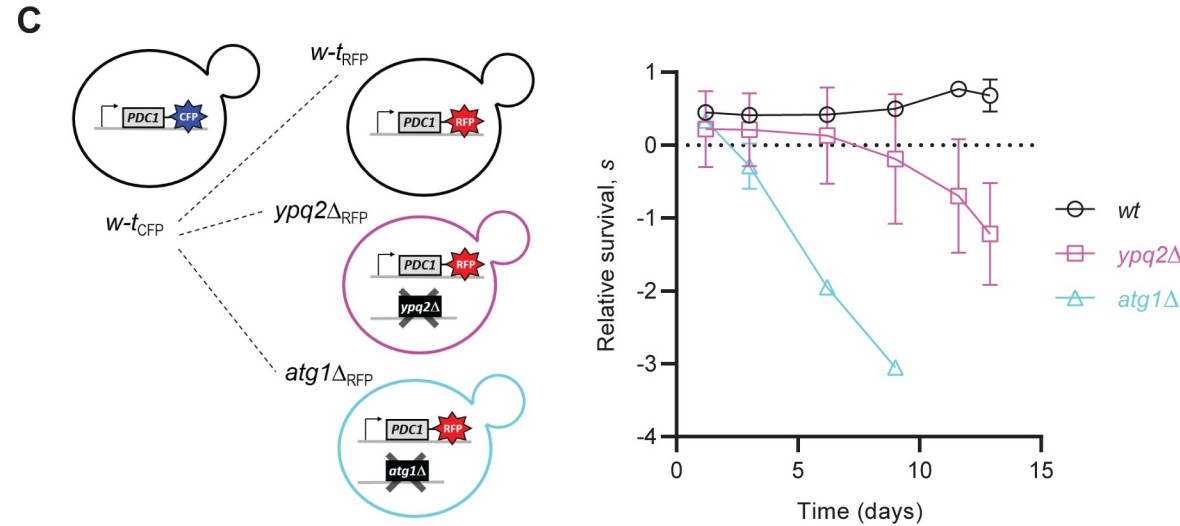

**Fig 4. Ypq2 is involved in mobilizing the intravacuolar Arg pool of nitrogen-starved cells.** *(A)* Percentage of the initial arginine pool left after 6 h of N starvation. The intracellular Arg content was measured in the *w-t*, *ypq2Δ*, *car1Δ*, and *ypq2Δ car1Δ* strains before and 6 hours after transfer to an N-free medium. Error bars represent the SD (**: $p < 0.005$; ****: $p < 0.0001$ by the Brown-Forsythe and Welch ANOVA tests). (n = 6–7). *(B)* [14]C-Arg export rates were measured from vesicles isolated from the *w-t* and *ypq2Δ* strains and loaded with [14]C-Arg (50 μM) as detailed in Materials and methods. Error bars represent the SD (*: $p < 0.05$ by paired t-test). (n = 4). *(C)* Competitive survival of RFP-labeled *w-t* or mutant strains relative to a CFP-labeled *w-t* reference, as a function of days after transfer to an N-free medium. The relative survival (*s*) at each sampling point is defined as the average background-corrected *ln*(RFP/CFP) interpolated at a fixed time point after outgrowth. Error bars are the SD of three gene-deletion replicates for the *w-t*/*w-t* and *ypq2Δ*/*w-t* competitions, while the *atg1Δ*/*w-t* control competition was carried out once.

[17,29]. Alternatively, Ypq2 could preferentially function as a uniporter. To assess this view, we determined whether Ypq2 can mediate Arg export in the absence of any extravacuolar compound. To this end, we isolated vacuoles from *w-t* and *ypq2Δ* cells, caused their hypo-osmotic rupture, and the broken vacuoles were then reconstituted into vesicles in the presence of [14]C-Arg (50 μM). The vesicles filled with radiolabeled Arg were then collected and used to measure initial rates of Arg export (Fig 4B). The values obtained with vesicles from the *ypq2Δ* strain were ~2-fold lower compared to those from the *w-t*. This result indicates that Ypq2 can function as a uniporter catalyzing net Arg export. The residual export detected in the absence of Ypq2 further shows that Arg efflux can occur independently of Ypq2, which is in line with the observation that Arg is still consumed at a significant rate in N-starved *ypq2Δ* mutant cells (Fig 4A). In other words, at least one additional vacuolar transporter most likely contributes to intravacuolar Arg export.

To further investigate the role of Ypq2 in N-starved cells, we assessed the survival of *ypq2Δ* mutant cells under N-starvation conditions. Specifically, we outgrew fluorescent-protein labeled *ypq2Δ* mutant cells in competition with *w-t* reference cells, and measured the changes in relative fluorescence with time after N starvation (see Methods). We found that the relative survival of the *ypq2Δ* strain was initially similar to that of the *w-t*, but significantly compromised after 11 days after N starvation (Fig 4C; *p* < 0.05, t-test). The autophagy-impaired *atg1Δ* control strain showed a rapid and severe drop in relative survival, as expected. Impaired survival of the *ypq2Δ* mutant suggests that Ypq2 is required to increase survival of N-starved cells, most likely by making stored vacuolar Arg available as N source.

All in all, these results indicate that a major function of Ypq2 is to catalyze net efflux of Arg stored in the vacuole when cells face N starvation, and that lack of such activity is particularly detrimental to survival during prolonged transfer under N starvation.

## Vsb1 is a novel vacuolar transporter-like protein essential to arginine storage in the vacuole

In N-replete *w-t* cells, Arg is stored at high concentration in the vacuole, where it binds to polyphosphates. Our next aim was to identify the transporters involved in this Arg import. Ypq2 is unlikely to play this role, as intracellular Arg is not reduced in *ypq2Δ* mutant cells (Fig 5A). We even detected with good reproducibility a significantly higher total Arg content in the *ypq2Δ* mutant, a phenotype more consistent with a role of Ypq2 in exporting intravacuolar Arg. This phenotype contrasts with that of the *vtc4Δ* mutant, unable to produce vacuolar polyphosphates [20,30]: in *vtc4Δ* cells, intracellular Arg is much reduced (Fig 5A), confirming that vacuolar polyphosphates strongly influence the Arg storage capacity of the vacuole.

According to a previous study based on measuring transport into reconstituted vacuolar vesicles, the Vba1, Vba2, and Vba3 transporters are involved in uptake of basic amino acids into the vacuole and likely function as $H^+$ antiporters [13,28]. We did not observe, however, any reduction of total cellular Arg in *vba1-2-3Δ* cells (Fig 5A). Furthermore, vacuoles isolated from *vba1-2-3Δ* cells showed the same Arg uptake rate as *w-t* cells (Fig 5B). These results

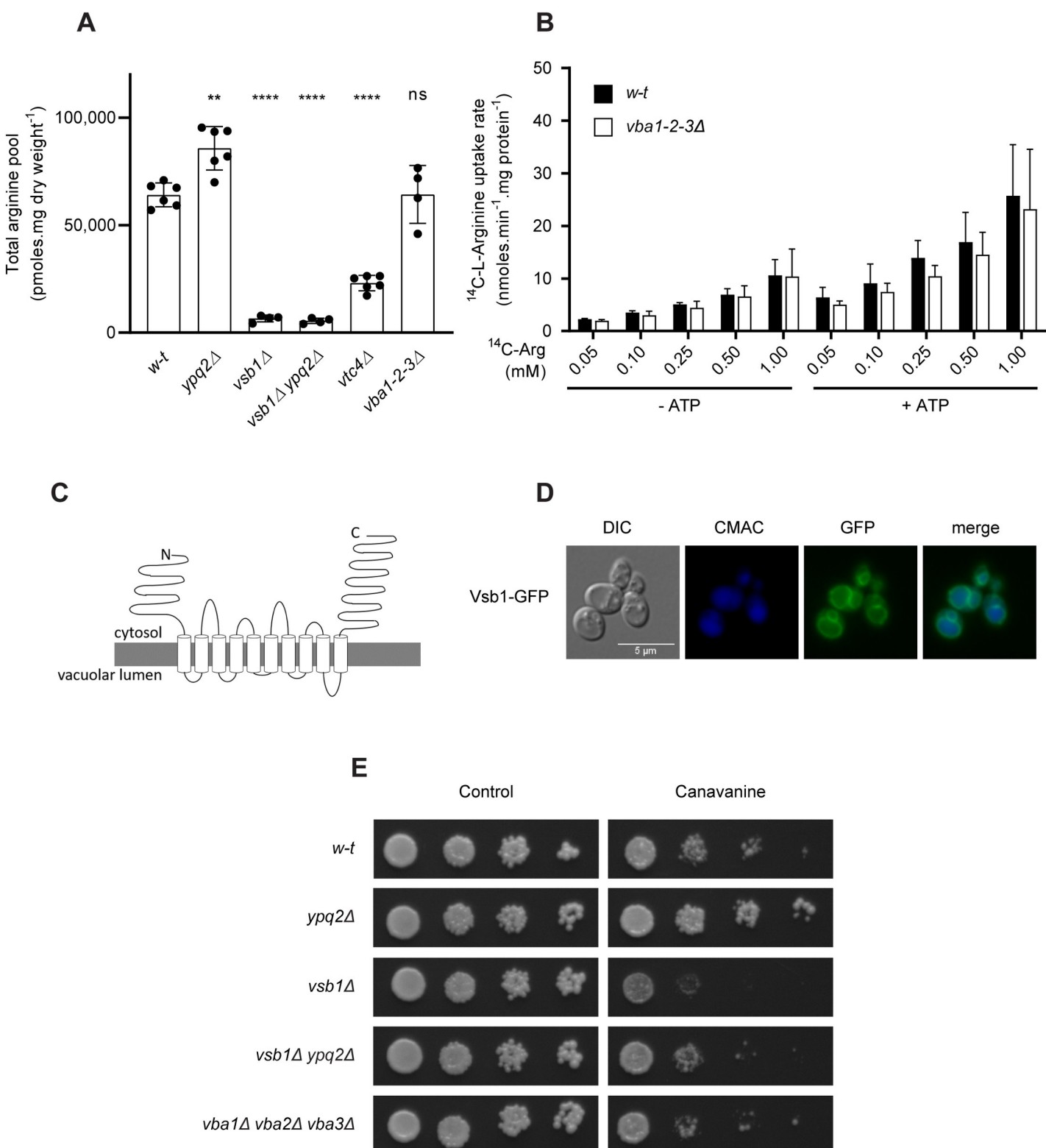

**Fig 5. Vsb1/Ygr125w is essential to storage of Arg in the vacuole.** *(A)* The intracellular Arg content was measured in the *w-t*, *ypq2Δ*, *vsb1Δ*, *ypq2Δ vsb1Δ*, and *vtc4Δ* strains (\*\*: $p < 0.005$ and \*\*\*\*: $p < 0.0001$ by the Brown-Forsythe and Welch ANOVA tests) (n = 4–6). *(B)* Accumulation of $^{14}$C-Arg added at different concentrations (mM) was measured, after a 4-minute incubation, in vacuoles isolated from the *w-t* and *vba1-2-3Δ* strains (n = 3). The vacuoles were incubated for 8 minutes in the absence or presence of ATP (4 mM) before addition of $^{14}$C-Arg. *(C)* Schematic representation of the Vsb1 protein inserted into the vacuolar membrane. *(D)* Vsb1

localizes to the vacuolar membrane. Yeast cells transformed with a low-copy-number plasmid expressing the *VSB1-GFP* fusion gene under the natural *VSB1* promoter were grown on minimal glucose $NH_4^+$ medium, and analyzed by epifluorescence microscopy. CMAC was used to label the vacuolar lumen in blue. *(E)* The *vsb1Δ* mutant is hypersensitive to canavanine, and this phenotype is epistatic over the canavanine resistance phenotype of the *ypq2Δ* mutant. Cells spread on minimal glucose $NH_4^+$ medium containing canavanine (0.6 μg/ml) or not were incubated for 3 days at 29°C.

suggest that in cells growing on minimal glucose $NH_4^+$ medium, the Vba1-2-3 proteins do not play an important role in Arg accumulation in the vacuole, which means that other transport proteins are involved.

Mülleder et al. [31] have measured the amino acid content in 4,913 yeast gene-deletion mutants. The cells used for this analysis were prototrophic for all amino acids and grown on a minimal glucose $NH_4^+$ medium, as under our working conditions. The *ypq2Δ* mutant analyzed in this study displayed a slightly elevated cellular Arg content, in keeping with our data. We thus analyzed the Arg content data of the remaining 4,912 strains, retrieved a list of mutants displaying a reduced Arg level, and crossed it with the list of 303 yeast membrane transporters inventoried in the Yeast Transport Protein database [32]. Our attention was drawn to gene *YGR125W*, as its deletion results in one of the strongest reductions in the cellular Arg pool. The His and Lys contents were also reduced in this mutant, but those of most of the other amino acids were not [31]. *YGR125W* encodes a protein belonging to the APC (amino acid-polyamine-organocation) superfamily of transporters and distantly related in sequence to the yeast Sul1 and Sul2 sulfate permeases [33], but of unknown biological function. The protein contains 10 predicted transmembrane segments flanked by unusually large N- and C-tails (Fig 5C). Furthermore, a previous proteomic analysis of isolated vacuoles revealed enrichment of vacuolar fraction in this protein [34]. We thus hypothesized that Ygr125w might be involved in transport of Arg into the vacuole and tentatively named it Vsb1 (for Vacuolar Storage of Basic amino acids 1). To test this hypothesis, the gene encoding GFP-fused Vsb1 was expressed in *w-t* cells under the *VSB1* promoter. The fusion protein was found to localize to the vacuolar membrane (Fig 5D). We then analyzed the total cellular Arg pool in a *vsb1Δ* mutant strain and found it to be much lower than that of the *w-t* (Fig 5A). The *vsb1Δ* mutant also displayed high hypersensitivity to canavanine (Fig 5E), a phenotype opposite to the canavanine resistance of the *ypq2Δ* mutant. This hypersensitivity is likely due to failure to sequester the toxic Arg analogue in the vacuole. Furthermore, analysis of the *vsb1Δ ypq2Δ* strain revealed that the low Arg pool of the *vsb1Δ* mutant is epistatic over the higher Arg pool of the *ypq2Δ* mutant (Fig 5A). The double *vsb1Δ ypq2Δ* strain also displayed intermediate sensitivity to canavanine, as compared to the *vsb1Δ* and *ypq2Δ* single mutants. These results suggest that Vsb1 is essential to Arg storage in the vacuole.

## No Vsb1-dependent arginine transport activity is detectable in isolated vacuoles

We next sought to detect Vsb1-dependent uptake of Arg into isolated vacuoles. Such activity might, for instance, correspond to the low-affinity Arg uptake detected with vacuoles from *ypq1-2-3Δ* mutant cells at high Arg concentration (Fig 2D). We thus first examined this possibility. The residual Arg uptake displayed by vacuoles from the *ypq1-2-3Δ* triple mutant was unaffected by additional deletion of the *VSB1* gene (Fig 6A). In other words, no Vsb1-associated Arg uptake activity was detected in the triple mutant, even at high Arg concentration. We next compared Arg uptake by vacuoles isolated from *w-t*, *vsb1Δ*, and *ypq2Δ* cells. We unexpectedly found the *vsb1Δ* and *ypq2Δ* mutants to exhibit the same phenotype, namely a strong reduction of Arg uptake (Fig 6B). A possible interpretation of this observation is that the Vsb1-dependent activity is undetectable in our *in vitro* conditions and Ypq2 is inactive in

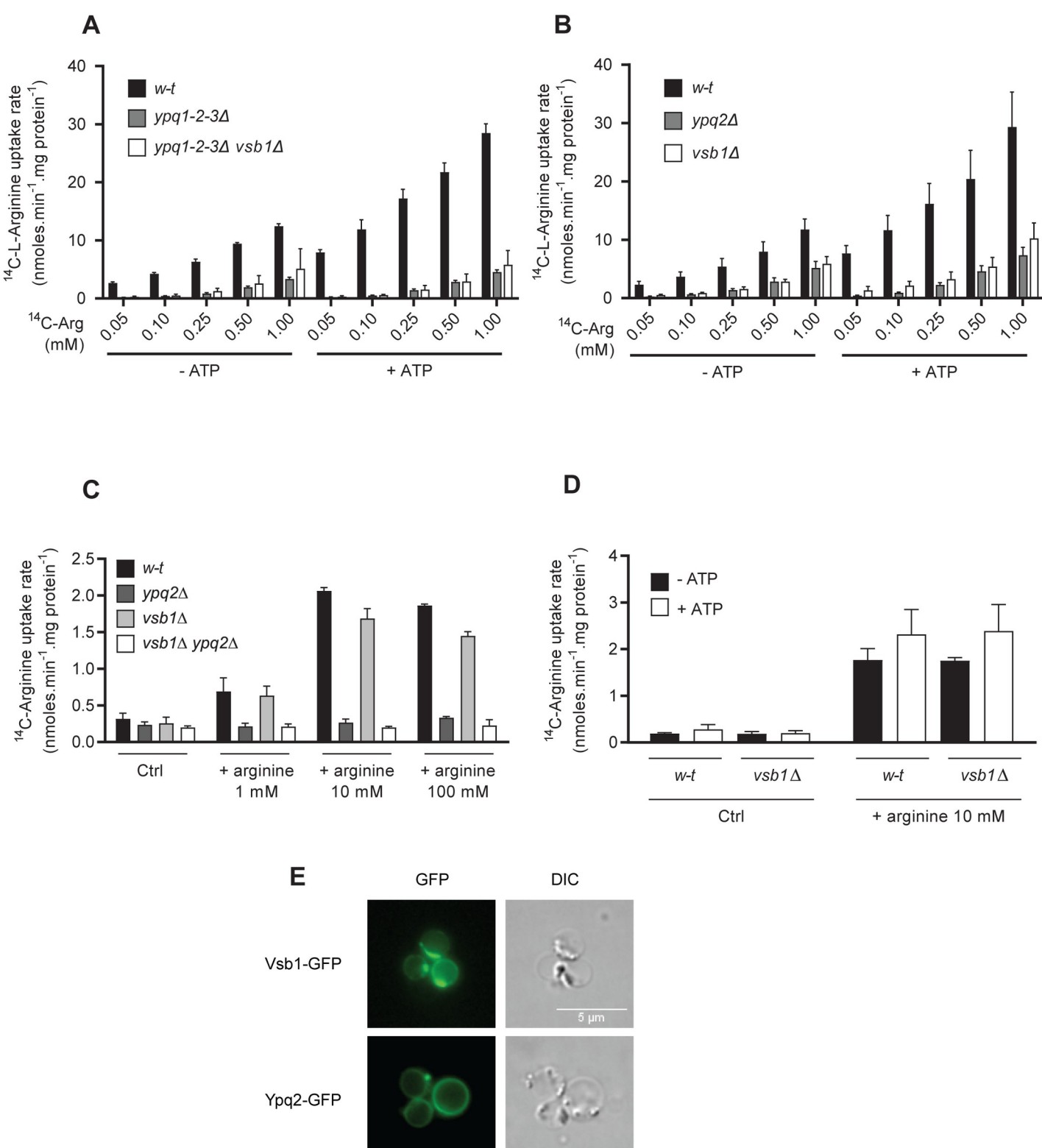

**Fig 6. No Vsb1-associated Arg uptake is detectable in isolated vacuoles.** *(A)* The accumulation of $^{14}$C-Arg added at different concentrations (mM) was measured, after a 4-minute incubation, in vacuoles isolated from the *w-t*, *ypq1-2-3Δ*, and *ypq1-2-3Δ vsb1Δ* strains. The vacuoles were incubated for 8 minutes in the absence or presence of ATP (4 mM) before addition of $^{14}$C-Arg (n = 2–3). *(B)* Same as in (A) except that the strains were the *w-t*, *ypq2Δ*, and *vsb1Δ* strains (n = 3–4). *(C)* The uptake of $^{14}$C-Arg added at 50 μM was measured, after a 4-minute incubation, in reconstituted vacuolar vesicles isolated from the *w-t*, *vsb1Δ*, *ypq2Δ*, and *vsb1Δ ypq2Δ*

strains. Vesicles were filled with a buffer without Arg (Ctrl) or containing 1 mM, 10 mM, or 100 mM non-radiolabeled Arg. (n = 2). *(D)* The uptake of $^{14}$C-Arg added at 50 μM was measured, after a 4-minute incubation, in reconstituted vacuolar vesicles isolated from the *w-t* and *vsb1Δ* strains. Vesicles filled with a buffer without Arg (Ctrl) or containing 10 mM non-radiolabeled Arg were incubated for 8 minutes in the absence or presence of ATP (4 mM) before addition of $^{14}$C-Arg. (n = 2). *(E)* Localization of Vsb1-GFP and Ypq2-GFP in isolated vacuoles analyzed by epifluorescence microscopy. For all experiments, error bars represent the SD.

*vsb1Δ* mutant cells because the intravacuolar Arg concentration is too low to drive $^{14}$C-Arg uptake via the Arg/Arg exchange reaction described above. To test this view, we isolated intact vacuoles from *vsb1Δ* mutant cells and, after rupture, reconstituted vacuolar vesicles in the presence of different concentrations of unlabeled Arg (Fig 6C). Remarkably, loading with Arg restored a high level of Arg uptake by vesicles from the *vsb1Δ* mutant, and this restored activity depended on Ypq2. Hence, the lack of Ypq2 activity displayed by vacuoles isolated from the *vsb1Δ* mutant is an indirect effect of the low luminal Arg content of these vacuoles. Furthermore, contrarily to our expectations, neither isolated vacuoles (Fig 6A and 6B) nor reconstituted vesicles (Fig 6C and 6D) showed any Vsb1-associated Arg uptake, whether ATP was present or not. The main Arg uptake activity detectable in our *in vitro* assay is thus that of Ypq2, and Vsb1 is inactive or nearly so under these conditions. We hypothesized that this lack of Vsb1 activity might result from rapid downregulation of the protein (due to targeting to the vacuolar lumen, for example) caused by the harsh treatment to which cells are subjected before spheroplast lysis. Yet Vsb1-GFP was still detected at the membrane of isolated intact vacuoles, albeit at a lower level than Ypq2-GFP (Fig 6E). We also noticed that a large fraction of the Vsb1-GFP was clustered in subregions of the vacuolar membrane, a pattern not observed in whole cells. Overall, these results suggest that the *in vitro* conditions used to measure Avt1 and Ypq2 transport activity are not suitable for detecting that of Vsb1. Alternatively, Vsb1-associated transport activity might be inhibited via a regulatory mechanism, perhaps related to the apparent clustering of the protein. For instance, this negative control might be induced during the harsh treatment to which cells are subjected before vacuole isolation.

## Vsb1 is required for arginine uptake into the vacuoles of growing cells

Our failure to detect Vsb1-associated Arg uptake into isolated vacuoles or derived vesicles prompted us to assess *in vivo* whether Vsb1 might play a role in Arg uptake into vacuoles. For this we compared growing *w-t* and *vsb1Δ* mutant cells. We first measured the accumulation of $^{14}$C-Arg in whole cells, reasoning that overall accumulation of external Arg might be altered if Arg accumulation in the vacuole is impaired. We found *w-t* and *vsb1Δ* cells to show similar initial uptake rates (Table 1), indicating that Can1, the main plasma-membrane Arg permease

**Table 1. Initial uptake rate of $^{14}$C-Arg in whole cells.**

| Strain | $^{14}$C-Arg initial uptake rate (nmoles.min$^{-1}$.mg protein$^{-1}$) |
|---|---|
| *w-t* | 21,3 ± 1,4 |
| *w-t* + bafilomycin A | 26,3 ± 3,1 |
| *vba1Δ vba2Δ vba3Δ* | 19,9 ± 4,5 |
| *ypq2Δ* | 25,3 ± 1,1 |
| *vsb1Δ* | 18,0 ± 2,1 |

Strains were grown on minimal ammonium medium and uptake at 250 μM was measured after 20, 40, 60 and 80 seconds of incubation. Values were then used to calculate the initial uptake rate via linear regression (GraphPad Prism) (n = 3). Additionally, a *w-t* strain was incubated for 15 minutes with bafilomycin A (1 μM) before starting the uptake.

active under the growth conditions used, is similarly active in the two strains. Long-term accumulation of $^{14}$C-Arg, however, was strongly reduced in the *vsb1Δ* mutant (Fig 7A), a phenotype not observed with the *vba1-2-3Δ* mutant (Fig 7B).

We next sought to better understand why long-term accumulation of Arg is impaired in the *vsb1Δ* mutant despite a normal initial uptake rate. We have previously shown that Can1, upon catalysis of Arg transport, undergoes ubiquitylation, endocytosis, and targeting to the vacuole, where it is degraded. This downregulation is triggered in Can1 by a conformation switch that unmasks a cytosolic sequence targeted by Art1, an α-arrestin under the control of TORC1 (Target Of Rapamycin Complex 1) [35,36]. We reasoned that the Can1 downregulation induced by Arg transport might be accelerated in *vsb1Δ* mutant cells because the Arg entering them fails to enter the vacuole and thus accumulates in the cytosol. In support of this view, we found the *vsb1Δ* mutant to display much more pronounced Arg-elicited Can1-GFP endocytosis (Fig 7C). Thus, a lack of Vsb1 results in more efficient Can1 downregulation upon Arg uptake. Importantly, this observation also indicates that Vsb1 is active in growing cells, where it likely plays an important role in Arg sequestration within the vacuole.

To further test this view, we examined whether Vsb1 promotes accumulation in the vacuole of $^{14}$C-Arg having entered growing cells. If so, the *vsb1Δ* mutant should display more $^{14}$C-Arg in the cytosol and less in the vacuole. We reasoned that this could be highlighted by selectively permeabilizing the plasma membrane and measuring the amount of cytosolic $^{14}$C-Arg eluted after washing (Fig 7D). The amount of $^{14}$C-Arg present in the vacuole could then be released by treating the permeabilized cells with $H_2O$ to cause hypo-osmotic rupture of the organelle. We thus set up this assay, using cytochrome C (cytC) as a permeabilizing agent [37] and CMAC as a vacuolar dye to check the hypo-osmotic rupture of the vacuole (Figs 7D and S2). This method was used to compare *w-t* and *vsb1Δ* mutant cells having incorporated $^{14}$C-Arg. To make sure that a similar amount of $^{14}$C-Arg was internalized by both strains, we introduced into them an additional *npi1/rsp5* mutation impairing Can1 ubiquitylation and endocytosis [35], and when necessary we also adjusted the external $^{14}$C-Arg concentration. We also introduced into the strains a *car1Δ* mutation preventing arginase-mediated $^{14}$C-Arg catabolism. We found $^{14}$C-Arg having accumulated for 30 minutes in growing *w-t* cells to be equally distributed between the cytC and $H_2O$ eluates (Fig 7E). This was to be expected if part of the $^{14}$C-Arg having entered the cells was incorporated into the vacuole, although a contribution of other organelles to $^{14}$C-Arg sequestration and some level of cross-contamination cannot be ruled out. In addition, a small amount (5–15%) of the accumulated radiolabeled amino acid remained in the cells after the osmotic shock (S2 Fig). This effect is likely due at least in part to incorporation of $^{14}$C-Arg into proteins not released into the eluates. Importantly, *vsb1Δ* mutant cells displayed a much greater amount of $^{14}$C-Arg in the CytC eluate than in the $H_2O$ eluate (Fig 7E). This result, expected if a greater amount of $^{14}$C-Arg remained in the cytosol, indicates that Vsb1 contributes importantly, in growing cells, to sequestration of $^{14}$C-Arg in the vacuole. As a control experiment, we incubated cells with $^{14}$C-Glutamate (Glt). This amino acid is known to be largely present in the cytosol, where it can be converted to glutamine [17]. Upon addition of $^{14}$C-Glt, a much larger fraction of $^{14}$C-labeled compounds was recovered in the CytC eluate, and importantly, the *vsb1Δ* mutation had no effect on their distribution between the cytC and $H_2O$ eluates (Fig 7F).

As Vsb1-dependent sequestration of $^{14}$C-Arg within the vacuole can be detected with this assay, we next investigated whether this sequestration is sensitive to inhibitors of the V-ATPase. Remarkably, when *w-t* cells were treated with bafilomycin A before $^{14}$C-Arg accumulation, more radioactivity was recovered in the cytC eluate than released after $H_2O$ treatment (Fig 7G), as in untreated *vsb1Δ* mutant cells (Fig 7E). This result is compatible with the model that the activity of Vsb1 depends on the $H^+$ gradient established by the V-ATPase at the

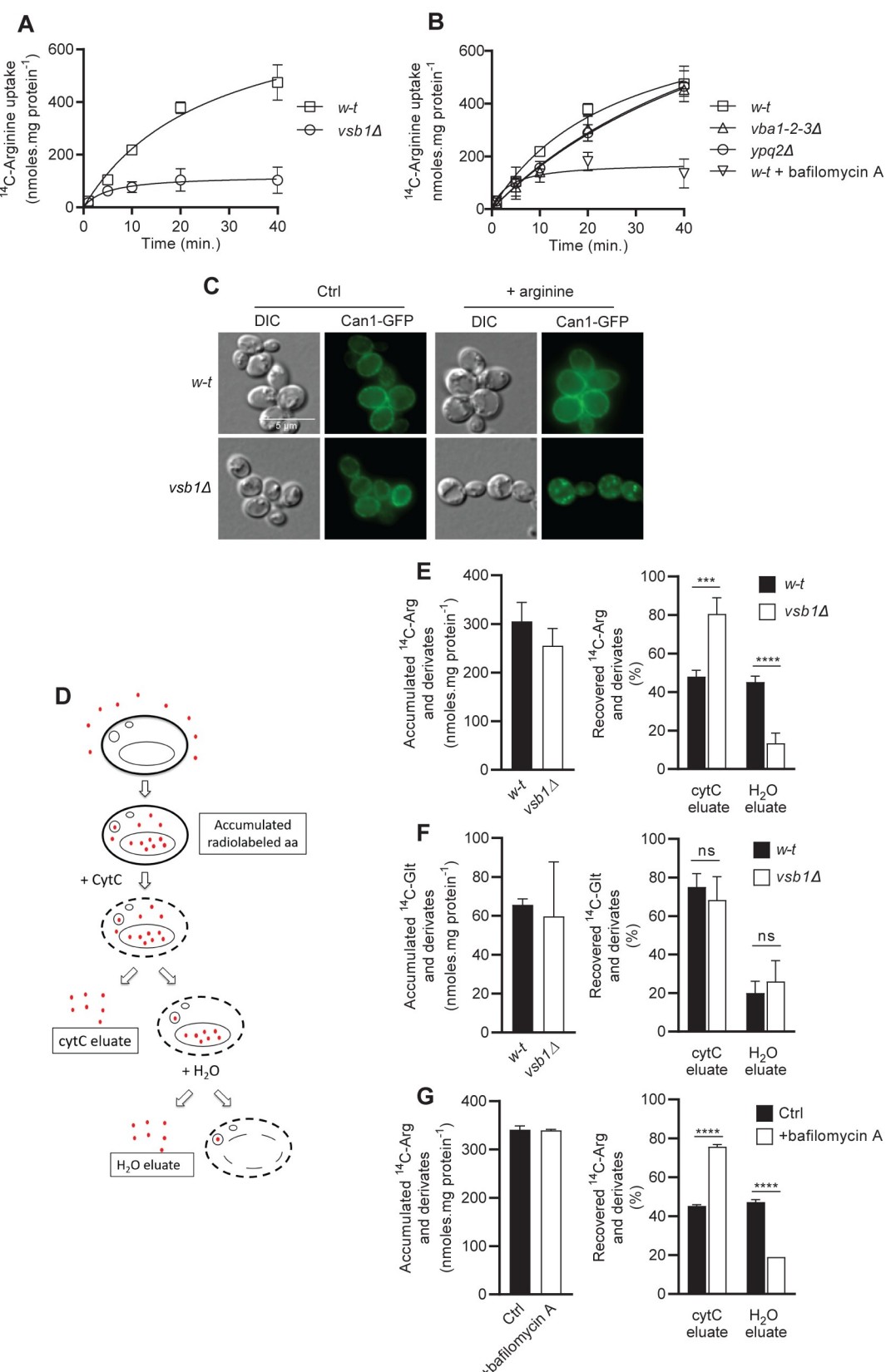

**Fig 7. Vsb1 promotes arginine uptake into the vacuoles of growing cells.** *(A)* Time-course accumulation of $^{14}$C-Arg (250 μM) in whole cells of the *w-t* and *vsb1Δ* strains over 40 min. *(B)* Same as in A, except that cells were of the *w-t*, *vba1-2-3Δ*, and *ypq2Δ* strains and of the *w-t* strain treated with bafilomycin A (1 μM) (n = 2). *(C)* Microscopy analysis of Can1-GFP localization in the *w-t* and *vsb1Δ* strains before and 15 min after Arg addition (5 mM). *(D)* Schematic representation of the experimental setup for cell permeabilization. After incubation with the radiolabeled amino acid and measurement of the accumulated radioactivity, cells were permeabilized by treatment for 1 h with cytochrome C, and the eluted radioactive content (cytC eluate) was counted. Permeabilized cells were then treated with $H_2O$ to cause an osmotic shock releasing the vacuolar content ($H_2O$ eluate). *(E)* The $^{14}$C-Arg accumulated in *w-t* (SL073) and *vsb1Δ* (SL074) cells was measured after a 10-min incubation (38 μM and 150 μM $^{14}$C-Arg, respectively) *(left)*. After cell permeabilization, the distribution of initially accumulated $^{14}$C between the cytC and $H_2O$ eluates was determined *(right)* (n = 4; ***: p<0.005 and ****: p<0.001 by Student's t-test). *(F)* The $^{14}$C-Glt accumulated in *w-t* and *vsb1Δ* cells was measured after a 10-min incubation (150 μM $^{14}$C-glutamate) *(left)*. After cell permeabilization, the distribution of initially accumulated $^{14}$C between the cytC and $H_2O$ eluates was determined *(right)* (n = 4; ns: p>0.05 by Student's t-test). *(G)* The $^{14}$C-Arg accumulated in untreated (Ctrl) and bafilomycin-treated *w-t* cells was measured after a 10-min incubation (50 μM and 100 μM $^{14}$C-Arg, respectively) *(left)*. After cell permeabilization, the distribution of initially accumulated $^{14}$C between the cytC and $H_2O$ eluates was determined *(right)* (n = 4; ****: p<0.001 by Student's t-test). For all experiments, error bars represent the SD.

vacuolar membrane. To further assess this view, we examined the influence of V-ATPase inhibition on $^{14}$C-Arg uptake into whole cells. We observed a normal initial uptake but a strongly reduced long-term accumulation of the radiolabeled amino acid, a phenotype again close to that of the *vsb1Δ* mutant (Fig 7B).

All in all, these results indicate that Vsb1 is active at the vacuolar membrane of growing cells, where it is necessary for Arg uptake. Furthermore, this uptake is presumably driven by the $H^+$ gradient established by the V-ATPase.

## Vsb1 and Ypq2 seem inversely regulated by nitrogen

The above results show that cells lacking Vsb1 display several phenotypes indicative of an important role of this protein in active transport of Arg into the vacuole. Yet we detected no Vsb1-dependent uptake of Arg into isolated vacuoles, the main Arg transport activity detected in these vacuoles being due to Ypq2. This transporter is active in N-starved cells, where it plays an important role in mobilizing intravacuolar Arg. These observations prompted us to hypothesize that Vsb1 and Ypq2 might be regulated inversely. Specifically, the Vsb1-associated transport, detectable in cells growing on N-rich medium, might be inhibited upon a shift to N starvation and also in cells subjected to the harsh treatment preceding the isolation of vacuoles. Conversely, Ypq2 might be inactive in cells growing on N-rich medium and become active upon N starvation and during the treatment to which cells are subjected before vacuole isolation.

To assess this model, cells growing on N-rich medium were transferred for 3 h to a medium devoid of N and subsequently incubated with $^{14}$C-Arg. The cytC permeabilization assay was then applied to determine the distribution of internalized $^{14}$C-Arg (Fig 8A). During initial growth in the presence of N, *w-t* cells used as a control again showed equal distribution of internalized $^{14}$C-Arg between the cytC and $H_2O$ eluates. When the same cells were N-starved, the cytC eluate was enriched in internalized $^{14}$C-Arg. The latter $^{14}$C-Arg distribution was in fact similar to that displayed by the *vsb1Δ* mutant, whether the medium contained N or not: ~80% of the $^{14}$C-Arg was found in the cytC eluate. These results suggest that Vsb1 is inhibited upon a shift to N starvation. As this inhibition was not associated with any detectable relocation of Vsb1 to the vacuolar lumen (Fig 8B), it would appear to affect the intrinsic activity of the protein. We next sought to detect Ypq2 activity in N-replete growing cells. We first observed that, contrarily to *vsb1Δ* cells, whole cells of the *ypq2Δ* strain showed no reduction of $^{14}$C-Arg accumulation under these conditions (Fig 7B). When we applied to these cells the cytC permeabilization assay, we observed no significant influence of Ypq2 on the distribution

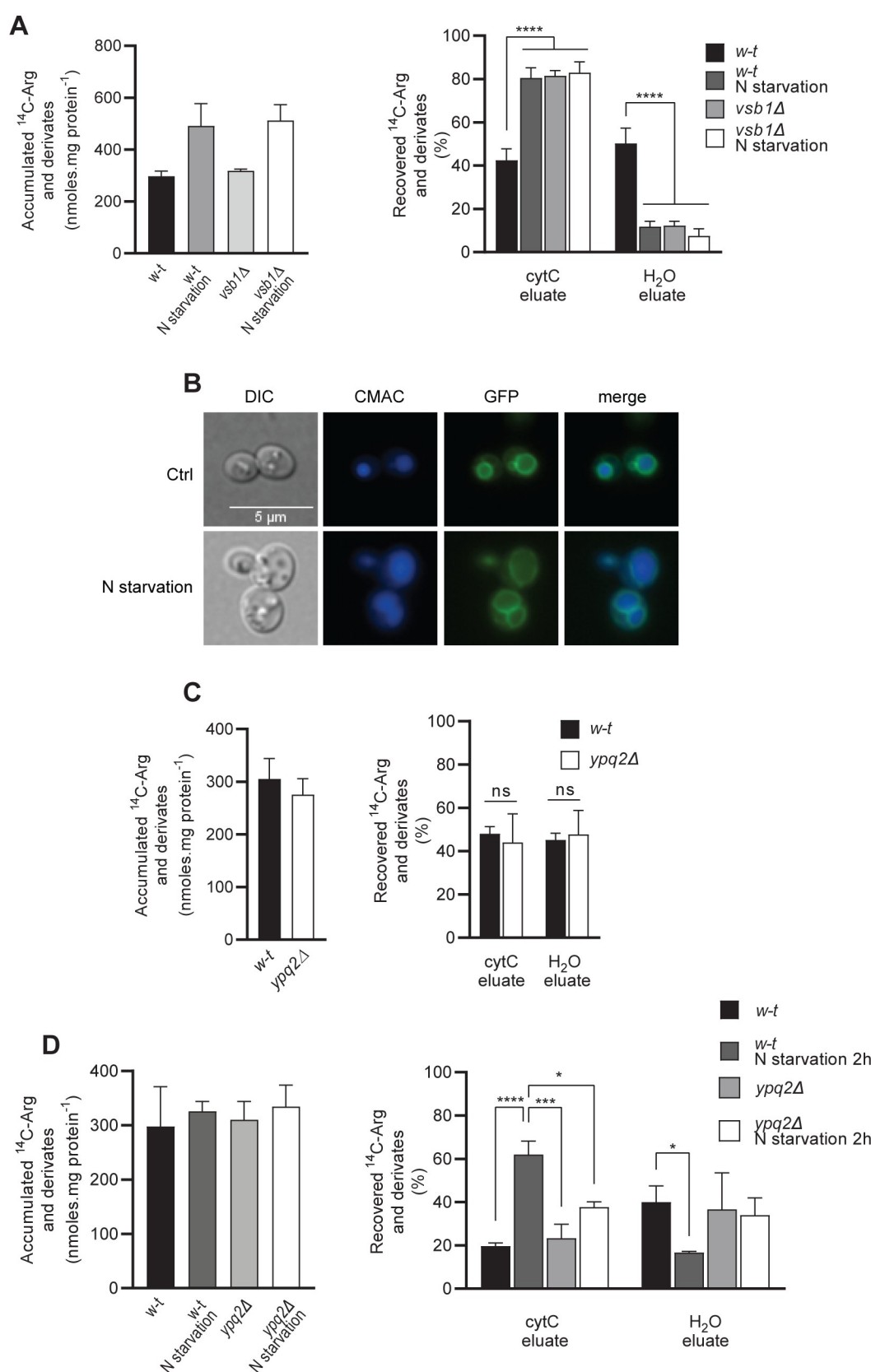

**Fig 8. Vsb1 and Ypq2 seem inversely regulated by nitrogen.** *(A)* Cells of the *w-t* and *vsb1Δ* mutant strains were grown with $NH_4^+$ as sole N source. Half of each culture was transferred to N-free medium for 3 h and subsequently incubated for 10 min in the presence of $^{14}$C-Arg added at 38 μM (*w-t* under ctrl conditions), 50 μM (*w-t* under N starvation), or 150 μM (*vsb1Δ* mutant under both conditions) (*left*). After cell permeabilization, the distribution of initially accumulated 14C between the cytC and $H_2O$ eluates was determined (*right*) (n = 4; ****: p<0.001 by one-way ANOVA). *(B)* Vsb1 remains located at the vacuolar membrane during N starvation. Cells transformed with a low-copy plasmid expressing the *VSB1-GFP* fusion gene under the natural *VSB1* promoter were grown on minimal glucose $NH_4^+$ medium (Ctrl) before transfer to N-free medium for 3 h and examination by epifluorescence microscopy. *(C)* The $^{14}$C-Arg accumulated in *w-t* and *ypq2Δ* cells was measured after a 10-min incubation (150 μM $^{14}$C-Arg) (*left*). After cell permeabilization, the distribution of initially accumulated $^{14}$C between the cytC and $H_2O$ eluates was determined (*right*) (n = 4; ns: p>0.05 by Student's t-test). *(D)* Cells of the *w-t* and *ypq2Δ* strains were incubated in the presence of $^{14}$C-Arg (200 μM) for 30 min. Half of each culture was then transferred to N-free medium for 2 h (*left*). After cell permeabilization, the distribution of initially accumulated $^{14}$C between the cytC and $H_2O$ eluates was determined (*right*). Exceptionally cells from a *CAR1* background strain were used in this experiment as the *car1Δ* mutation presumably inhibits Arg mobilization. (n = 3; *: p<0.05; ***: p<0.001; ****: p<0.001 by one-way ANOVA).

of $^{14}$C-Arg between the cytC and $H_2O$ eluates (Fig 8C), although the protein could in principle exchange $^{14}$C-Arg in the cytosol with unlabeled Arg inside the vacuole. These results suggest that Ypq2 is not active in N-replete cells, possibly because it undergoes regulatory inhibition. As Ypq2 is present at the vacuolar membrane under these conditions [5], we would expect this putative inhibition of Ypq2 to target the protein's intrinsic activity.

If Ypq2 is inhibited in N-replete cells, this negative control should be relieved upon a shift to N starvation, as the protein contributes importantly to consumption of stored Arg under these conditions. To see if vacuolar Arg is indeed exported via Ypq2 upon N starvation, cells were incubated for 30 min in the presence of $^{14}$C-Arg before transfer to an N-free medium for 2 h. The cytC permeabilization assay was then applied to these cells and to control cells not starved of N (Fig 8D). In the *w-t* strain grown in the presence of N, the major part of the free radioactivity was recovered in the $H_2O$ eluate, as expected if $^{14}$C-Arg is mainly stored in the vacuole. The phenotype of *ypq2Δ* mutant cells was similar, again illustrating that Ypq2 does not contribute to vacuolar sequestration of Arg. The situation was different for *w-t* cells starved of N for 2 h: a large fraction of the internalized $^{14}$C-Arg (~60%) was released into the cytC eluate. This result confirms that the vacuolar Arg stores are mobilized during N starvation. In contrast, N starvation of *ypq2Δ* mutant cells did not significantly alter the distribution of $^{14}$C-Arg between the cytC and $H_2O$ eluates. This indicates that the radiolabeled amino acid was trapped in the vacuole. These results are in line with a model where, contrarily to Vsb1, Ypq2 is active under N starvation, allowing the cell to mobilize and consume its vacuolar Arg stores.

In conclusion, these observations show that Vsb1 is active in N replete cells and inactive upon N starvation, whereas the opposite seems to hold for Ypq2. This suggests that Ypq2 and Vsb1 are inversely regulated according to the N supply conditions. Furthermore, the vacuoles isolated from cells and used for uptake assays might be equivalent to those of N-starved cells. This might explain why Ypq2 instead of Vsb1 shows up as the most active Arg transport system in these *in vitro* experiments.

## Discussion

The present study aimed to identify and characterize the transporters involved in import and export of Arg across the yeast vacuolar membrane and to determine whether their activities could be coordinated according to the N supply conditions. Our first step was to implement methods for isolating intact vacuoles and measuring the activity of vacuolar transporters [38]. We have also developed a method, based on plasma membrane permeabilization followed by osmotic rupture of the vacuole, for assessing *in vivo* the contribution of a transporter to

sequestration of a compound in the vacuole or to its export from the organelle. As discussed below, this method proved particularly useful, as our data suggest that a vacuolar transporter active *in vivo* might be inactivated during the harsh treatment to which cells are subjected during the experimental steps preceding vacuole isolation.

One of the main conclusions of this work is that in N-replete cells, Vsb1 is essential to Arg accumulation in the vacuole (Fig 9). Lack of this vacuolar transmembrane protein causes drastic reduction of intracellular Arg [31], a result confirmed under the working conditions used here. The luminal Arg concentration of vacuoles isolated from *vsb1Δ* mutant cells is reduced to the extent that it can no longer efficiently drive the exchange reaction catalyzed by the Ypq2

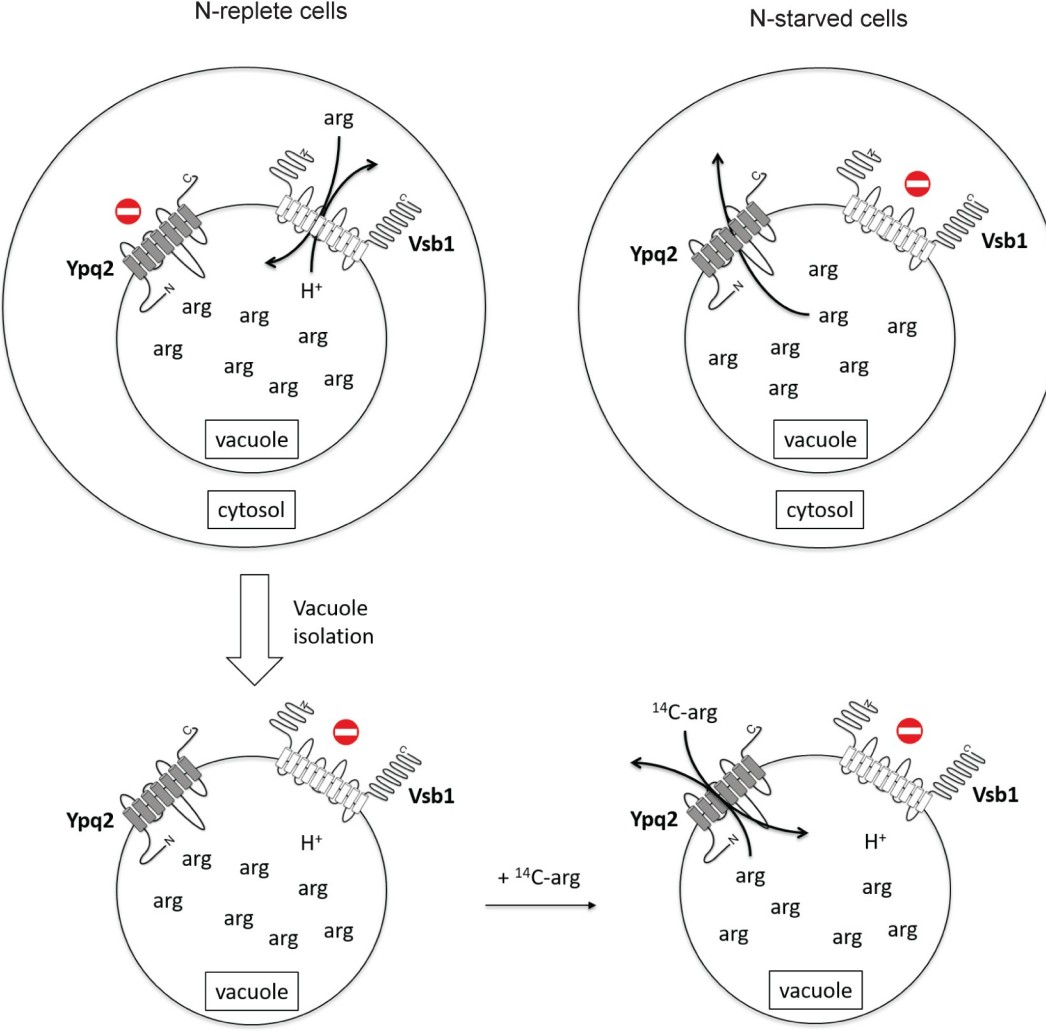

**Fig 9. Model for the coordinated import and export of Arg across the vacuolar membrane.** In N-replete cells, Vsb1 is active and promotes Arg uptake into the vacuole, probably through a proton antiport mechanism. The Ypq2 export protein is maintained inactive so that Arg is sequestered and accumulates in the vacuole. When cells are transferred to an N-free medium, Vsb1 is inactivated while Ypq2, which becomes active, mobilizes the intravacuolar Arg to the cytosol, where it is used as an N source. This Arg export likely corresponds to a uniport reaction. During the procedure for isolating vacuoles, cells pregrown under N-replete conditions undergo harsh conditions causing inactivation of Vsb1 and concomitant activation of Ypq2, as occurs under N starvation conditions. Measured uptake of $^{14}$C-Arg in those vacuoles is thus mainly mediated by Ypq2, catalyzing a $^{14}$C-Arg/Arg exchange reaction.

transporter (see below). Furthermore, we confirmed by cell permeabilization experiments that Vsb1 is required to sequester radiolabeled Arg in the vacuole just after its uptake into growing cells. That this accumulation was impaired upon inhibition of the V-ATPase suggests that Vsb1-dependent activity corresponds to an Arg/H$^+$ antiport, although indirect effects caused by this inhibition cannot be ruled out. The *vsb1Δ* mutant exhibits additional phenotypes consistent with a role of Vsb1 in Arg transport into the vacuole. For instance, it is hypersensitive to canavanine, probably because the toxic analog fails to enter the vacuole. The intracellular pools of His and Lys are also much reduced in the *vsb1Δ* mutant, whereas those of other amino acids remain largely unaltered (34). Thus, Vsb1 is likely involved in the transport of all three canonical cationic amino acids. Yet further work, including purification and reconstitution into proteoliposomes, is needed to confirm that Vsb1 is a secondary active transporter of these three amino acids. For instance, we cannot rule out the possibility that Vsb1 affects Arg transport only indirectly, by acting as a positive regulator of Arg importers.

Our work also suggests an unexpected property of Vsb1. While its activity can be measured *in vivo* in N-replete cells, it is lost upon a shift to N starvation, suggesting that Vsb1 might be inhibited (Fig 9). As the Vsb1-GFP protein remained present at the vacuolar membrane under N starvation, this putative Vsb1 inhibition seems to concern the intrinsic activity of the protein rather than its location, in contrast to the recently demonstrated targeting of some vacuolar transporters to the vacuolar lumen [39–41]. In addition, we failed to detect any Vsb1-dependent Arg uptake activity *in vitro*, using either intact vacuoles or reconstituted vesicles. Like Ypq2-GFP, Vsb1-GFP localizes to the limiting membrane of these vacuoles, where a fraction of it seems to cluster in domains of unknown nature. This particular lateral distribution has not been observed for Ypq2-GFP, nor even for Vsb1-GFP when detected in whole cells. Why no Vsb1-associated activity was detectable *in vitro* remains unclear. The conditions used in the uptake assay might be inappropriate for detecting Vsb1 activity, but this seems unlikely, as our assay is suitable for measuring Avt1-catalyzed H$^+$-coupled tyrosine uptake. This prompts us to propose that the putative negative regulation affecting Vsb1 upon N starvation might also cause Vsb1 inhibition during the harsh treatments to which cells are subjected prior to vacuole isolation (Fig 9).

Our study has also characterized Ypq2 as a transporter that contributes importantly to export of intravacuolar Arg under N starvation (Fig 9). We find that initial intracellular Arg stocks, known to be mostly vacuolar in N-replete cells, are not properly consumed via cytosolic arginase after a shift to N starvation if the shifted cells lack the Ypq2 protein. Furthermore, the results of our cell permeabilization and *in vitro* assays fully support a direct role of Ypq2 in net export of intravacuolar Arg upon N starvation. A role of Ypq2 in exporting intravacuolar Arg is further supported by the canavanine resistance displayed by the *ypq2Δ* mutant, which is most likely due to increased sequestration of the toxic analog in the vacuole. Although Ypq2 is clearly involved in mobilizing vacuolar Arg for use as an N source, our results also support the view that additional transporters contribute to export of intravacuolar Arg under N starvation conditions.

Ypq2 is very similar to the human lysosomal transporter PQLC2, shown to catalyze export of cationic amino acids from *Xenopus* oocytes when targeted to their plasma membrane [5]. Interestingly, it has also been established that LAAT-1, the homolog of PQLC2 and Ypq2 in *Caenorhabditis elegans*, is important in supplying Arg and Lys to the cytosol during embryogenesis. This LAAT-1-catalyzed transport was abolished when the conserved proline residue in the first PQ-loop of the protein was mutated to leucine [42]. A similar effect was reported for Ypq2 when an alanine was substituted for the corresponding residue (Pro29), highlighting the close similarity between PQLC2, LAAT-1 and Ypq2, and further indicating that Ypq2, like its orthologs, is an Arg transporter [9]. As for Vsb1, however, reconstitution of the purified

protein in proteoliposomes is needed to fully characterize the transport activity of Ypq2. The simplest model supported by our data is that export of intravacuolar Arg catalyzed by Yqp2 is not coupled to $H^+$ efflux and is a uniport reaction (Fig 9). In support of this view, Ypq2-mediated Arg import and export were detected in vesicles without prior establishment of an $H^+$ gradient by the V-ATPase. Yet the measured activity of Ypq2 is about twice as high if the V-ATPase is active. This suggests that to function optimally, Ypq2 requires an acidic lumen, in keeping with its role in export from the vacuole. Our results also reveal that the Arg import associated with Ypq2 *in vitro* is mainly an Arg/Arg exchange (Fig 9). This is illustrated by our experiments in which reconstituted vacuolar vesicles were loaded with unlabeled Arg before performing the $^{14}$C-Arg uptake assays. These observations are fully compatible with the properties of the vacuolar Arg transport system previously described by Wiemken and collaborators [12,27]. By a considerable feat in experimentation, these authors were able to show that Arg uptake is in fact coupled to its export, with 1:1 stoichiometry. Although such an Arg/Arg exchange reaction is counterintuitive at first glance, counterflow has been previously shown to occur for some transporters when the substrate is present on both sides of the membrane [43]. We thus propose that upon a shift to N starvation, the Arg concentration in the cytosol rapidly drops, thereby establishing a steep gradient across the vacuolar membrane. Ypq2 would then catalyze net efflux of Arg via a uniport reaction. Once the Arg concentration in the cytosol reaches values close to the apparent $K_m$ of the protein, Ypq2 could then catalyze an exchange reaction. Interestingly PQLC2, formerly known as "system C", was first described as an exchanger [44,45]. Hence, Ypq2 and PQLC2 seem to display remarkable similarities. The above model of Ypq2 function, however, must also take into account another observation suggesting that this transporter, like Vsb1, might be under N control. Specifically, although Ypq2 clearly mediates the export of intravacuolar Arg under N starvation (when Vsb1 is inactive), we failed to detect in N-replete cells any contribution of Ypq2 to transport of Arg into the vacuole, even though, in principle, the Arg/Arg exchange activity of Ypq2 should be measurable under such conditions. This suggests that Ypq2 is inactive in N-replete cells, probably to avoid futile export of Arg entering the vacuole via Vsb1 (Fig 9). As Ypq2 localizes to the vacuolar membrane in N-replete cells, its putative inhibition would occur at the protein activity level.

Further work is needed to assess the validity of the model that both Vsb1 and Ypq2 are under N regulation, to dissect the molecular mechanisms underlying these proposed regulations, and to identify the signaling pathways that orchestrate them according to the N supply conditions. It will also be interesting to determine whether the proposed activation of Ypq2 under starvation applies also to human PQLC2. This view seems compatible with the observation that the ability of PQLC2 to recruit the C9orf72 complex is stimulated in response to cationic amino acid starvation [46]. Further investigation of Vsb1 and Ypq2 regulation may also shed new light on the coordinated regulation of lysosomal/vacuolar transporters in general. Lastly, we propose that the methodology set up in this study can be used to identify novel vacuolar nutrient transporters and investigating their regulation.

## Materials and methods

### Yeast strains, plasmids, and growth conditions

The reference wild-type (23344c) and the mutant strains used in this study (Table 2) derive from the wild-type strain Σ1278b [47], except where otherwise stated. Cells were grown at 29°C on a minimal buffered medium, pH 6.1 [48] with glucose (3%) as a carbon source and ammonium in the form of $(NH_4)_2SO_4$ (10 mM) as the nitrogen source, except where stated otherwise. The plasmids used in this study are listed in Table 3.

**Table 2. Strains used in this study.**

| Strain | Genotype | Reference |
|---|---|---|
| 23344c | *ura3* | Laboratory collection |
| EL031 | *ura3 ypq2Δ* | [5] |
| LL180 | *ura3 ypq1Δ ypq2Δ ypq3Δ* | [5] |
| SL017 | *ura3 uga1::Sna3-pHluorin* | This study |
| SL015 | *ura3 uga1::Sna3-pHluorin Sna4-dsRed* | This study |
| COM035 | *ura3 avt1Δ* | This study |
| COM075 | *ura3 vtc4Δ* | This study |
| LL162 | *ura3 vba1Δ vba2Δ vba3Δ* | This study |
| COM090 | *ura3 vsb1Δ* | This study |
| SL025 | *ura3 ypq1Δ ypq2Δ ypq3Δ vsb1Δ* | This study |
| SL073 | *ura3 npi1 car1Δ* | This study |
| SL074 | *ura3 npi1 car1Δ vsb1Δ* | This study |
| SL075 | *ura3 npi1 car1Δ ypq2Δ* | This study |
| SL076 | *ura3 npi1 car1Δ vsb1Δ ypq2Δ* | This study |
| YEG011 | *PDC1-Cerulean his3Δ::kanMX4* | [49] |
| YEG012 | *PDC1-mCherry his3Δ::kanMX4* | [49] |
| YEG4555 | *PDC1-mCherry atg1Δ::kanMX4* | [49] |
| JUC009 | *PDC1-mCherry ypq2Δ::hphMX4* | This study |

## Measurement of total amino acid pools

Yeast cells were grown to mid-exponential phase. The dry weight was measured on 25-ml cell culture ($0.4^{*}10^{7}$ cells/ml) filtrates dried at 60°C for 24 h. Another 25 ml of the same culture was washed three times in $H_2O$. Cells were resuspended in 2 ml $H_2O$ and boiled for 15 min. To collect condensation drops and remove cell debris, the extract was centrifuged (13,000 g for 1 min) and filtered (0.45 μm, Millipore). The resulting extracts were subjected to amino acid analysis with the AccQ Tag Ultra Derivatization Kit (Waters) as described previously [51].

## Isolation of vacuoles and vesicles

Intact vacuoles were prepared as follows, on the basis of a previously described protocol [25,38]. One-liter cultures of cells growing in minimal medium were harvested at mid-exponential phase. The cells were resuspended in 25 ml of 0.1 M Tris-HCl pH 8.9, 10 mM DTT, incubated for 10 min at 29°C, and collected by centrifugation. Zymolyase digestion was carried out for 30 min at 30°C in 15 ml digestion mixture (50 mM potassium phosphate pH 7.5, 600 mM sorbitol in minimal medium with 0.3% glucose and 1 mM PMSF, 370U zymolyase).

**Table 3. Plasmids used in this study.**

| Plasmid | Description | Reference |
|---|---|---|
| pSL081 | YCpVSB1-GFP (URA3) | This study |
| pSL073 | YCP-UGA1 RECI-G418R-TPIp-SNA3-pHluorin-UGA2t-UGA1 RECII (URA3) | This study |
| pSL069 | YCP-UGA1 RECI-G418R-TPIp-SNA3-pHluorin-UGA2t-PGK1p-SNA4-Ds-Red- UGA1 RECII (URA3) | This study |
| pKG036 | YCpCAN1-GFP (URA3) | [50] |
| pLL161 | YCpYPQ2-GFP (URA3) | [5] |

Spheroplasts were collected by centrifugation and gently lysed with 7mg/l DEAE-dextran in 15 ml PS buffer (10 mM PIPES/KOH pH 6.8, 200 mM sorbitol) 15% Ficoll 400, first for 2 min at 0˚C and then for 2 min at 30˚C. The samples were overlaid with 2.5 ml of 8% Ficoll 400, 3.5 ml of 4% Ficoll 400, and 1.5 ml of 0% Ficoll 400 (in PS buffer). After ultracentrifugation (150,000 g; 90 min, 4˚C), vacuoles were harvested from the 0–4% interface. For experiments with reconstituted vesicles, the vacuole suspensions were successively diluted 2x in 20 mM PIPES/KOH pH 6.8 and 2x in 10 mM PIPES/KOH pH 6.8, incubated for 5 min on ice, and collected by centrifugation (13,000 g; 15 min; 4˚C). The vesicles were then resuspended in PS buffer. A protease inhibitor cocktail (100 mM pefabloc SC, 100 ng/ml leupeptin, 50 mM 1,10-phenanthroline (Merck; catalogue number: 131377) and 500 ng/ml pepstatin A) was included in all vacuole buffers after spheroplasting. Vacuole and vesicle amounts were estimated by the protein content, determined using the Bradford assay (BioRad).

### Fluorescence microscopy

Growing cells were laid down on a thin layer of 1% agarose, and isolated vacuoles were diluted 10x in PS buffer. Both were viewed at room temperature with an epifluorescence microscope (Eclipse E600; Nikon) equipped with a 100× differential interference contrast N.A. 1.40 Plan-Apochromat objective (Nikon) and appropriate filters. Images were captured with a digital camera (DXM1200; Nikon) and ACT-1 acquisition software (Nikon) and were processed with Fiji software [52]. In each figure, we typically show only a few cells, representative of the whole population. Labeling of the vacuolar lumen with CMAC was performed by adding the fluorescent dye to a concentration of 25 μM at least 30 min prior to visualization. Labeling of the vacuolar membrane of whole cells with FM4-64 was performed as described previously [53]. Labeling of the membrane of isolated vacuoles with FM4-64 was performed by adding the dye to 16 μM to a sample of isolated vacuoles.

### In vitro measurement of intravacuolar pH

Intact vacuoles (20 μg) expressing Sna3-pHluorin were diluted in 100 μl assay buffer (PS buffer supplemented with 150 mM KCl and 4 mM MgCl$_2$). Fluorescence intensities at 395 and 475 nm were measured with a microplate reader (Synergy Mx BioTek) with a 512/9 nm emission filter and 395/9 and 475/9 nm excitation filters. The I395 nm to I475 nm emission intensity ratio was used to calculate the vacuolar pH. Vacuoles isolated from a *w-t* strain not expressing pHluorin were used to measure background fluorescence. The calibration curve was generated as described previously [54–56].

### Uptake assays in whole cells

Accumulation of $^{14}$C-labeled amino acids in whole cells was measured at the indicated time points in whole-cell uptake assays, as described previously [38,50]. The measured counts were normalized to protein concentration. Transport was expressed in nmol/mg protein per unit of time and reported as mean values (the number of biological replicates is indicated in each figure), with error bars representing standard deviations (SD).

### Cell permeabilization assays

After a 10-min whole-cell uptake assay, 5 ml culture was filtered and washed with H$_2$O. For plasma membrane permeabilization, cells were resuspended in 4 ml cytochrome C solution (1mg/ml in sorbitol 1M) and incubated for 1 hour at 4˚C with gentle shaking. The cell suspension was then percolated over a glass microfiber filter (GF/C 25 mm) and washed 3 times with

1 ml of 1M sorbitol. Both the flow-through and the washing eluates were collected as the "cytC eluate". Vacuoles were permeabilized by addition of $H_2O$ (2x3 ml) and the flow-through was collected as the "$H_2O$ eluate". All cell membranes left intact were disrupted by adding 3 ml of 50% methanol followed by a final 3-ml wash with $H_2O$. The eluate was collected as the "Methanol eluate". All eluates (1 ml each) and the filters with their lingering cells were mixed with 3 ml scintillation liquid (Ultima-Flo AP), except for the "cytC eluate", a 500-μl sample of which was mixed with 12 ml scintillation liquid. The accumulated counts were measured with a scintillation counter (Beckman Coulter LS 6500 Liquid Scintillation Counter). Eluate counts are normalized to the counts of the initial culture and displayed as means ±SD.

## Uptake assays on vacuoles and vesicles

As described in [38], 20 μg isolated intact vacuoles or 30 μg of reconstituted vacuolar vesicles were diluted in 100 μl assay buffer. Vacuoles were incubated for 8 min at 27˚C in a water bath with or without ATP (4 mM). The uptake reaction was started by adding the radiolabeled amino acid and stopped at the indicated time points by diluting the reaction medium in 500 μl cold assay buffer and putting the samples on ice. Depending on the conditions, the indicated inhibitors were added before or together with ATP. Similarly, competitors were added at the same time as the radiolabeled chemical. After uptake, vacuoles were centrifuged with a soft spin (5,000 g, 5 min, 4˚C). Once the supernatant was discarded, 500 μl cold assay buffer was added again and the vacuoles were centrifuged with a hard spin (13,000 g, 3 min, 4˚C). The supernatant was discarded once more and the vacuoles resuspended in 1 ml scintillation liquid before the accumulated counts were measured. Transport was expressed in nmol/mg protein per unit of time and reported as the mean of 2 biological replicates ±SEM. Initial uptake rates (30 seconds) at 8 substrate concentrations spread over a relevant range (2.5–250 μM) were used to determine the apparent Michaelis constant ($K_m$) and maximal velocity ($V_{max}$) for $^{14}C$-Arg uptake. The values were well fitted by a single-site Michaelis-Menten model.

## Export assays on vesicles

Vesicles were prepared as described above except that they were resuspended in PS buffer containing 50 μM $^{14}C$-Arg. After a second centrifugation (13,000 g; 15 min; 4˚C), $^{14}C$-Arg loaded-vesicles were resuspended in 100 μl assay buffer [38] and incubated at 27˚C. At each time point, 500 μl ice-cold assay buffer was added to the vesicles, which were then collected by centrifugation (5,000 g, 5 min, 4˚C), resuspended in 500 μl cold assay buffer, and collected again by centrifugation (13,000 g, 3 min, 4˚C). The supernatant was discarded once more and the vesicles resuspended in 1 ml scintillation liquid before the accumulated counts were measured. As the yields of vacuole rupture and $^{14}C$-Arg-filled vesicle formation substantially varied according to samples, the export rates were not calculated per mg of protein but as the number of nanomoles of $^{14}C$-Arg lost per minute of incubation in a constant 100 μl assay volume. The values obtained for vesicles from *ypq2Δ* cells were expressed relatively to those from the wild-type vesicles from the same biological replicate.

## Competitive survival experiments

Isogenic Cerulean (CFP) and mCherry (RFP) labeled strains were grown separately overnight, mixed in a 1:1 volume ratio, and inoculated in minimal ammonium medium until mid-log phase (OD 0.6–0.8). Mixed cultures were washed twice in sterile water and resuspended in 10 mL N-free medium and kept at 30˚C and shaken at 200 rpm. Nitrogen-starved competition cultures were outgrown at different time points (days) by inoculating 15 μL aliquots into 150 μL low-fluorescence SC medium [49] in six technical replicates. At each sampling point,

outgrowth cultures were monitored every hour during 16 h in an automated robotic station (Tecan Freedom EVO200) by collecting data for both fluorescence channels (CFP: Ex 433 nm/ 5 nm and Em 475 nm/5 nm; RFP: Ex 587 nm/5 nm and Em 610 nm/5 nm) and optical density at 600 nm (OD) in a multilabel plate reader (Tecan Infinite M1000). The relative survival (*s*) at each sampling point is defined as the average background-corrected *ln*(RFP/CFP) interpolated at a fixed time point after outgrowth, as described [49].

### Reproducibility of experiments and statistics

Three or more biological were carried out for the majority of experiments. For experiments presenting data from two biological replicates, those of a third experiment have often been included for some of the tested conditions. Furthermore, preliminary tests to establish the conditions for carrying out the two illustrated experiments gave data fully consistent with the presented results. Statistical tests depending on normal distribution, standard deviation and number of samples were carried out using the GraphPad Prism software and are indicated in the figure legends. A P value of less than 0.05 was considered statistically significant.

### Supporting information

**S1 Fig. Ypq2 contributes to His uptake in isolated intact vacuoles.** *(A)* The accumulation, in vacuoles isolated from *w-t*, *ypq2Δ* and *ypq1-2-3Δ* strains, of $^{14}$C-L-His added at different concentrations (mM) was measured after a 4-minute incubation. The vacuoles were incubated for 8 minutes in the absence or presence of ATP (4 mM) before addition of $^{14}$C-His (n = 2). *(B)* Time course of $^{14}$C-Lys (100 μM) uptake into intact vacuoles from the *w-t* and *ypq2-3Δ* strains in the absence of ATP.
(TIF)

**S2 Fig. Permeabilization assay experiments.** *(A)* Schematic of the permeabilization assay protocol. After uptake of the radiolabeled compound, cells were permeabilized with cytochrome C and filtered. This allowed extraction of soluble amino acids and other metabolites present in the cytosol (cytC eluate). A subsequent osmotic shock with $H_2O$ led to permeabilization of the vacuolar membrane and release of the vacuolar content ($H_2O$ eluate). Any remaining intact cell membranes were then disrupted with 50% methanol (Methanol eluate). Lastly, the radioactivity contained in all eluates and the filter was measured. *(B)* Percentages of cells showing a CMAC-labeled cytoplasm or vacuole under control conditions (-cytC), after cytochrome C permeabilization of the plasma membrane (+cytC), and after permeabilization of the vacuole (+$H_2O$). *(C)* Fractions of the $^{14}$C-Arg and derivatives recovered during the permeabilization procedure. After uptake of $^{14}$C-Arg (38 μM), *w-t* cells underwent the full permeabilization procedure with either the permeabilization buffer containing cytochrome C or a control buffer with no cytochrome C. In the absence of cytochrome C in the buffer, most of the internalized radiolabeled compound was recovered in the Methanol eluate, suggesting that the cells were not permeabilized and that methanol effectively disrupted the cell membranes. In the presence of cytochrome C, most of the radioactivity was recovered in the cytC and $H_2O$ eluates and only a small fraction was eluted when methanol was added.
(TIF)

### Acknowledgments

We thank Bruno Gasnier for critical reading of the manuscript and regular exchanges, Andreas Mayer for useful discussions, Elie Saliba and Christos Gournas for advices, Pierre

Morsomme for the Sna4-DsRed and Sna3-GFP constructs, and Hana Sychrova and Olga Zimmermannova for the pHluorin gene construct.

## Author Contributions

**Conceptualization:** Melody Cools, Alexander DeLuna, Bruno André.

**Funding acquisition:** Bruno André.

**Investigation:** Melody Cools, Simon Lissoir, Elisabeth Bodo, Judith Ulloa-Calzonzin, Alexander DeLuna, Isabelle Georis.

**Methodology:** Melody Cools, Simon Lissoir, Elisabeth Bodo, Alexander DeLuna, Isabelle Georis.

**Project administration:** Melody Cools, Bruno André.

**Supervision:** Bruno André.

**Validation:** Melody Cools, Bruno André.

**Writing – original draft:** Melody Cools, Alexander DeLuna, Bruno André.

**Writing – review & editing:** Melody Cools, Bruno André.

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
