## [Decision Letter · Decision Letter 0]

14 May 2020

Dear Dr André,

Thank you very much for submitting your Research Article entitled 'Nitrogen coordinated import and export of arginine across the yeast vacuolar membrane' to PLOS Genetics. Your manuscript was fully evaluated at the editorial level and by independent peer reviewers. The reviewers appreciated the attention to an important topic but identified some aspects of the manuscript that should be improved.

We therefore ask you to modify the manuscript according to the review recommendations before we can consider your manuscript for acceptance. Your revisions should address the specific points made by each reviewer.

[LINK]

Yours sincerely,

Widmar Tanner

Guest Editor

PLOS Genetics

Gregory P. Copenhaver

Editor-in-Chief

PLOS Genetics

Concerning the aspects reviewer 2 raised as major concerns, it would suffice if the authors state more clearly that they present no direct evidence for the genes under discussion representing transport proteins.

Reviewer's Responses to Questions

**Comments to the Authors:**

Reviewer #1: This is a description of a well carried out study of how vacuoles help cells store and mobilize organic nitrogen. Two important gene-function relationships are identified, and kinetics and/or other important characteristics of their encoded transporters are described. Furthermore, the authors show the nitrogen status to have strong regulatory effects on the two transporters in a way that makes biological sense, although the mechanisms are yet obscure.

The manuscript is also well-written, creating a flow where generally the results from one experiment suggest questions that are addressed in the next one. This makes it easy and pleasant to read.

I have one general comment. The exchange function of Ypq2 is pointed out in a slightly wrong way. At places, the reader will think that Ypq2 has two states, one in which it is a uniporter and another where it exchanges molecules; that is in my view unlikely and would be a complicated assumption. All observations agree with the highly efficient exchange reaction being a side reaction that will always take place when there is Arg on both sides. It requires no energy, and there is no reason to think that it has any important biological consequence. Such exchange is also named counterflow and is a priori expected to be very commonly occurring. Let us assume that the function important for fitness in nature is uniport (export in the case of Ypq2) function and not exchange. All you need to explain the occurrence of a much higher rate of exchange than rate of net transport is a kinetic bottleneck in the conformational shift of the transporter without ligand between the inward-open conformation and the outward-open conformation. Such bottlenecks should a priori be expected to commonly arise in evolution on a seemingly random basis in various steps of transport cycles including the conformational shift mentioned, and indeed counterflow is often found in transporters when it is looked for, also when it has no apparent biological significance. Not only uniporters exhibit counterflow; good old LacY also has it (see e.g. Guan L, Kaback HR (2006) Lessons from lactose permease. Annu Rev Biophys Biomol Struct 35:67–91). I encourage you to state this or parts of it, perhaps in the Discussion. I have below indicated a few places in the manuscript where changes should be made with this view; there may be others. Nevertheless, I fully agree that the exchange function is an important part of the story that contributes to the flow guiding the reader.

I have many minor comments:

P1 Italicize “VSB1”.

Line 4 Consider changing “, and” into a semicolon.

15 Insert a comma after “nutrients” and another one after “arginine”, in both cases to indicate a non-defining relation.

76 Change “VTC” into “vacuolar transporter chaperone (VTC)”.

77 Change “mM)” into “mM phosphate groups)”.

103 Consider changing “seemed” into “appeared”.

125 I suggest changing “are” into “appear”, since the experiments described immediately above concern a single amino acid only.

136-140 First, you do not at this point of the story need to indicate one interpretation as being simpler than others. Second, if you wish to list possible explanations here, you need to mention a third, and in my mind even simpler, possible explanation, namely the one you reach later in the manuscript: [14C]Arg uptake can take place through Ypq2 by exchange with intravacuolar amino acid(s).

149 Change “w-t” into “w-t (23344c)” here and perhaps other places.

171 Change “Unsurprisingly, cold” into “As expected, non-radioactive” in order to make text understandable to readers unfamiliar to the convention “hot”-“cold”. Furthermore, change “neutral amino acids such as” into “the neutral amino acids”. I agree that it is likely that other neutral amino acids behave the same, but you should not conclude it unless of course you tried them, in which case you should describe it.

180 and 205 Insert “primarily” or “mainly” after “is”.

191 Change “previously described,” into “described for intact vacuoles,”.

199 Change “not.” into “not, although ATP stimulated the uptake.”.

204 Insert “into vacuole-derived vesicles” after “uptake”.

210-212 This is an overstatement. Instead, it would be correct to say “Appreciable Ypq2 activity does require, however, the presence of at least one of its substrates inside the vacuole, since little 14C-Arg was taken up by vacuolar vesicles devoid of Arg or His (Fig 3C).”, of course with “14” in superscript.

213-215 This is also an overstatement; it gives a wrong impression of the content of the three references given. None of the two first references mention a switch. In each case, a mitochondrial carrier is described which has a high kcat for exchange and a much lower yet significant kcat for uniporter activity. It is expected, and not a consequence of a shift of mechanism, that the relative significance of the latter becomes larger when the concentration ratio across the membrane increases. The third reference, on GLUT1, does describe a switch, but also in this case is the statement misleading. The effect of lowering the external glucose concentration has a large delay and occurs indirectly via the ATP/AMP energy system, a fact that in my view makes this reference less relevant; there is nothing that points to a similar mechanism for Ypq2. Most unprepared readers who do not read the three cited papers will erroneously think that the papers show cases where a high concentration ratio of a ligand (transport substrate) directly causes a change of transporter mechanism. This is not so; in references 30 and 31 one just sees a consequence of different experiments emphasizing different capabilities of the protein in question. The wording must be corrected, see my general comment and e.g. my comment to line 977.

223-225 Modify wording. I do not see that the hypothetical pH profile itself argues for the model, but it is indeed important to point out as an explanation for the effect of ATP, based on evolutionary considerations. I agree that, otherwise, readers might at this point think that the effect of ATP suggests that the pmf can drive transport through Ypq2, which I also agree appears unlikely.

261 Insert “than” after “more”.

297 I encourage you to reserve the name with SGD.

457 Consider changing “our observations show” into “we find”; the first could to some readers mean all observations described in the manuscript.

461 Consider changing “is” into “shows up as”.

533 Make “4” subscript.

595 “adopt” is misleading, see my general comment, and reconsider reference 32, see my comment to lines 213-215.

600 Also here, please rephrase slightly and avoid “switch”.

611 Insert “the” before “protein”.

657-659 The nomenclature for PS buffer must be clarified. Strictly interpreted, the layer e.g. immediately over the lysate should contain 15+8 percent Ficoll, but I think it only contains 8.

665 Change “O” to lower case, italic, since it means ortho; otherwise some readers will think it refers to binding to an oxygen atom. Another common name that you can choose is 1,10-phenanthroline. You can also add a vendor or producer and a catalog number.

669 and 996 Insert comma before “and”, since a new independent sentence starts.

691 Consider changing “Corrected for” into “normalized to”.

704-705 Specify the scintillation liquid. Counting efficiency depends on the identity of the scintillation liquid, the sample solvent, and the mixing ratio.

736 Change “m” into “nm”.

804 Delete “ (80-)”.

819 Change “enzymology” into “Enzymology”, and consider changing “177-96 p” into “pp 177-96”. Check full reference list for similar cases.

821 Change “Vacuolar” into “vacuolar” and “Cellular” into “cellular”. Check full reference list for similar cases.

831 Make the “+” superscript. Check full reference list for other cases of possible subscripts and superscripts.

955 Change “Arg uptake into” into “Arg uptake through Ypq2 into isolated”.

955-957 Specify in this experiment whether ATP was present or absent.

956-957 and 959 Change “ (see text for the strains used)” into “. Values for the ypq1-2-3Δ strain were subtracted from those for the ypq1-3Δ strain”, of course with the genotypes in italics.

958 Insert “of Ypq2” after “selectivity”.

963 Change “Vacuoles were” into “Vesicles were”.

965, 1049, 1052 and 1069 and Table 1 header: Make “14” superscript.

967, 971, 1007, 1010 and 1046 Change “ctrl” into “Ctrl”.

967, 971, 1008 and 1011 Consider changing “cold” into “non-radioactive”.

977 Change “Ypq2 switches into an exchange mode of transport” into “exchange will dominate, and net transport will slowly cease”.

1025 Change “w-t and vsb1Δ” into “w-t (SL073) and vsb1Δ (SL074)”.

1032 Change “ctrl” into “control (Ctrl)” or “untreated (Ctrl)”.

1085 Insert space between number and unit. Check full manuscript for similar cases.

Figures: In micrograph panels, either insert an indicated 10-micrometer bar (one per figure is sufficient) or indicate the length of sides of the panels.

Fig. 1A: Explain in the legend the scissors depicted in right panel.

Fig. 4A: Please delete this. The text is easy to understand by itself and can even be improved, and the two drawings make more confusion than explanation. There are at least two problems with them. First, the exchange reaction has no consequence on any concentrations; it is the uniporter function for both directions that assures that an equilibrium is approached. Second, the membrane potential and the negative charges of the polyphosphates make it unlikely that a cation like Arg will have the same concentration on both sides at equilibrium.

Fig. 4C: The ordinate axis of the right panel needs some explanation for non-experts on survival curves to understand. Move, e.g., the explanation on s from Materials and methods to the figure legend.

Fig.5A: Most genotype designations here end correctly with a delta, and that was intended also under the rightmost open bar. However, on my screen the character is erroneously a framed cross.

Fig. 5D: This is the first time the authors present CMAC. A brief explanation, at least of the purpose and principle, is needed in the legend.

Reviewer #2: The manuscript from Cools et al. focuses on the principal proteins that control arginine transport from and to the vacuoles in yeast, and provide evidence that this transport is regulated by nutrient requirements of the cell. The authors identify a novel vacuolar transmembrane protein, Vsb1, and provide compelling evidence that Vsb1 is as a major contributor to drive arginine transport into the vacuole. The authors also address the role of another transporter, Ypq2, a previously described exchanger of arginine (Boller et al Eur J Biochem. 1975) at the vacuole. The authors propose a model where Vsb1 imports arginine to the vacuole under nutrient rich conditions, and Ypq2 mediates the export to the cytosol during nitrogen starvation.

The authors present an overall convincing and elaborate study. They show by an elegant assay the contribution of each transporter to arginine uptake into the vacuole in vivo. Their method is based on the selective permeabilization of the plasma membrane followed by the osmotic rupture of the vacuole, thus allowing the recovery of the cytoplasmic and vacuolar pools of arginine.

Although the authors propose that Vsb1 is the main transporter of arginine import into the vacuole, they were not able to demonstrate Vsb1-dependent transport in vitro or its transport activity per se. Similarly, they propose that Ypq2 acts as an exporter (based on the deletion mutant analysis) during starvation conditions, yet they only show impaired uptake if the protein is deleted. Direct transport activity of these proteins by reconstitution was not shown, though also admitted by the authors. It is important to note that measurements in the Wiemken and collaborators study, as also in this work includes the whole set of transporters at the vacuole, so it is difficult to assign the observed exchange rates to only one transporter (as stated in the introduction, Lines 50-65)

My major concerns are listed below:

1. The authors claim that Ypq2 is a transporter (Lines 177-179); however, neither they nor others have shown direct activity of this protein, they probe activity in system (intact vacuoles) containing all or most of vacuolar protein. It would be important to stress this in their manuscript.

2. Along the same line, the authors show that Ypq2-dependent uptake is dependent on the intra-vacuolar concentration of arginine (Figure 3B and C) and propose that transport is bidirectional, and only unidirectional during starvation, although this was not assessed. Interestingly, they are able to detect activity even in the control condition that, although it is low, it is still stimulated by ATP. Can the authors rule out that the proposed exchange reaction is carried out by more than one transporter and that Ypq2 is rather the importer?

3. The authors propose that Ypq2 would mobilize Arg from the vacuole to the cytosol during nitrogen starvation (Figure 4). They convincingly show that Ypq2 is required for intracellular consumption of arginine during starvation; however, they do not show direct Ypq2-dependet export from the vacuole. To address this a bit more directly, the authors could measure the efflux of radioactive arginine that was previously incorporated into vacuoles before to demonstrate Ypq2-dependent export of this amino acid. Additionally, the export could be assessed under conditions of nutrient rich and starvation conditions.

4. The assumption that Ypq proteins are arginine transporters seem to be supported by the fact they homologues to the PLQ2 mammalian transporter. For PLQ2 it has been shown that mutation in P55 abolished arginine uptake. To demonstrate homologous activity, the authors could mutate the equivalent position in Ypq2 to test if the mutant is able or not to rescue the uptake phenotype they observe.

5. The authors claim that Vsb1 is a transporter, but do not show transporter activity. Although I do not expect them to demonstrate activity in vitro, I find it important that they remain cautious with their interpretation as also in their statements along the manuscript, as they refer to this protein as an essential transporter.

6. Figure 7: The analysis using bafilomicin A was performed in wt cells, so it is difficult to claim that the block in V-ATPase function specifically affects the activity of Vsb1. Please rephrase.

7. Along their entire manuscript the authors mix up a few things in my view. They observe under different conditions (N-depleted/repleted) different requirements of their transporters, and do so by comparing deletion mutants with wild-type. They then conclude that each of the transporters is regulated (in abstract and throughout the manuscript). However, they have no evidence at all that this is the case. Results from Figure 8 do not strictly show that these proteins are inactive in each condition, they are just not required. I agree that nitrogen regulation may be likely, but they do not show regulation of either transporter, and they should thus rephrase these passages accordingly.

8. I found the entire discussion too long and too excessive. The repeat many aspects of their results section. I recommend to condense it strongly (by ½) to make to more accessible.

Minor issues:

1. Line 124: “…the intact yeast vacuoles isolated via the above methods…”. The authors refer to the “above methods” as if they would explain them, but they do not describe the isolation method itself, so it would be suitable to say “our method” or “the used method”.

2 .Paragraph between lines 112-115 can be interpreted as they could not establish the pH measurement when incubated with the inhibitor. “ the pH measured in isolated vacuoles became more acidic in the presence of ATP (Fig 1), this acidification was partially lost upon addition of the K+/H+ ionophore nigericin, and it could not be established if the vacuoles were pre-incubated with the V-ATPase inhibitor bafilomycin A (Fig 1C)”

3. Figure 7C: It would be helpful for readers if the authors name of the protein fused to GFP (Can1) also in the figure.

4. Many experiments have been repeated only twice. Particularly, Figures 3B and 6D show the same control condition but the effect of ATP is only evidenced in Fig 3B. In this case as in other in which the n=2, at least 3 repeats should be included.

**Have all data underlying the figures and results presented in the manuscript been provided?**

Reviewer #1: Yes

Reviewer #2: Yes

PLOS authors have the option to publish the peer review history of their article (what does this mean?). If published, this will include your full peer review and any attached files.

Reviewer #1: Yes: Morten C. Kielland-Brandt

Reviewer #2: No

---

## [Editor Report · Decision Letter 1]

30 Jun 2020

Dear Dr André,

We are pleased to inform you that your manuscript entitled "Nitrogen coordinated import and export of arginine across the yeast vacuolar membrane" has been editorially accepted for publication in PLOS Genetics. Congratulations!

Yours sincerely,

Widmar Tanner

Guest Editor

PLOS Genetics

Gregory P. Copenhaver

Editor-in-Chief

PLOS Genetics

Comments from the reviewers (if applicable):

**Data Deposition**

http://datadryad.org/submit?journalID=pgenetics&manu=PGENETICS-D-20-00303R1

**Press Queries**

---

## [Editor Report · Acceptance letter]

4 Aug 2020

PGENETICS-D-20-00303R1 

Nitrogen coordinated import and export of arginine across the yeast vacuolar membrane 

Dear Dr André, 

We are pleased to inform you that your manuscript entitled "Nitrogen coordinated import and export of arginine across the yeast vacuolar membrane" has been formally accepted for publication in PLOS Genetics! Your manuscript is now with our production department and you will be notified of the publication date in due course.

With kind regards,

Kaitlin Butler

PLOS Genetics

On behalf of:
